# Structure-based discovery of fiber-binding compounds that reduce the cytotoxicity of amyloid beta

**Lin Jiang[†], Cong Liu[†], David Leibly, Meytal Landau[‡a], Minglei Zhao[‡b], Michael P Hughes, David S Eisenberg\***

Departments of Chemistry and Biochemistry and Biological Chemistry, Howard Hughes Medical Institute, UCLA–DOE Institute for Genomics and Proteomics, University of California, Los Angeles, Los Angeles, United States

**\*For correspondence:** david@mbi.ucla.edu

[†]These authors contributed equally to this work

[‡]**Present address:** [a]Department of Biology, Technion-Israel Institute of Technology, Haifa, Israel; [b]Department of Molecular and Cellular Physiology, Stanford University, Stanford, United States

**Competing interests:** The authors declare that no competing interests exist.

**Reviewing editor**: John Kuriyan, Howard Hughes Medical Institute, University of California, Berkeley, United States

**Abstract** Amyloid protein aggregates are associated with dozens of devastating diseases including Alzheimer's, Parkinson's, ALS, and diabetes type 2. While structure-based discovery of compounds has been effective in combating numerous infectious and metabolic diseases, ignorance of amyloid structure has hindered similar approaches to amyloid disease. Here we show that knowledge of the atomic structure of one of the adhesive, steric-zipper segments of the amyloid-beta (Aβ) protein of Alzheimer's disease, when coupled with computational methods, identifies eight diverse but mainly flat compounds and three compound derivatives that reduce Aβ cytotoxicity against mammalian cells by up to 90%. Although these compounds bind to Aβ fibers, they do not reduce fiber formation of Aβ. Structure-activity relationship studies of the fiber-binding compounds and their derivatives suggest that compound binding increases fiber stability and decreases fiber toxicity, perhaps by shifting the equilibrium of Aβ from oligomers to fibers.

## Introduction

Protein aggregates, both amyloid fibers and smaller amyloid oligomers, have been implicated in the pathology of Alzheimer's and other neurodegeneration diseases (*Chiti and Dobson, 2006*; *Eisenberg and Jucker, 2012*). The increasing prevalence of Alzheimer's disease in our aging societies, the associated tragedy for patients and their families, and the mounting economic burden for governments have all stimulated intense research into chemical interventions for this condition. Much work has been focused on screening compounds that prevent aggregation and the associated cytotoxicity of the amyloid β-peptide (Aβ) (reviews by *Sacchettini and Kelly, 2002*; *Bartolini and Andrisano, 2010*; *Hard and Lendel, 2012*).

Screens have often focused on natural products from plants and lichens. These include polyphenols, such as epigallocatechin gallate (EGCG) from green tea (*Ehrnhoefer et al., 2008*) and curcumin from the spice turmeric (*Yang et al., 2005*). These natural polyphenolic compounds show inhibition on the fibrillation of a variety of amyloid proteins, including Aβ40 as well as α-synuclein, IAPP and PrP (*Porat et al., 2006*; *Dasilva et al., 2010*; *Ono et al., 2012*). Several dyes have also been found to ameliorate amyloid toxicity. Orcein from lichens appears to diminish toxic oligomers and enhance fiber formation (*Bieschke et al., 2011*). Congo red, thioflavin T and their analogs, commonly used as staining reagents for amyloid detection, exhibit ameliorative effects on neurodegenerative disorders, such as Alzheimer's, Parkinson's, Huntington's, and prion diseases (*Frid et al., 2007*; *Alavez et al., 2011*), however their application is limited by significant side effects (*Klunk et al., 2004*).

Additional screens have identified a variety of molecules, including proteins (*Evans et al., 2006*), antibodies (*Kayed et al., 2003*; *Ladiwala et al., 2012*), synthetic peptide mimetics (*Findeis, 2002*;

**eLife digest** Alzheimer's disease is the most common form of dementia, estimated to affect roughly five million people in the United States, and its incidence is steadily increasing as the population ages. A pathological hallmark of Alzheimer's disease is the presence in the brain of aggregates of two proteins: tangles of a protein called tau; and fibers and smaller units (oligomers) of a peptide called amyloid beta.

Many attempts have been made to screen libraries of natural and synthetic compounds to identify substances that might prevent the aggregation and toxicity of amyloid. Such studies revealed that polyphenols found in green tea and in the spice turmeric can inhibit the formation of amyloid fibrils. Moreover, a number of dyes reduce the toxic effects of amyloid on cells, although significant side effects prevent these from being used as drugs.

Structure-based drug design, in which the structure of a target protein is used to help identify compounds that will interact with it, has been used to generate therapeutic agents for a number of diseases. Here, Jiang et al. report the first application of this technique in the hunt for compounds that inhibit the cytotoxicity of amyloid beta. Using the known atomic structure of the protein in complex with a dye, Jiang et al. performed a computational screen of 18,000 compounds in search of those that are likely to bind effectively.

The compounds that showed the strongest predicted binding were then tested for their ability to interfere with the aggregation of amyloid beta and to protect cells grown in culture from its toxic effects. Compounds that reduced toxicity did not reduce the abundance of protein aggregates, but they appear to increase the stability of fibrils. This is consistent with other evidence suggesting that small, soluble forms (oligomers) of amyloid beta that break free from the fibrils may be the toxic agent in Alzheimer's disease, rather than the fibrils themselves.

In addition to uncovering compounds with therapeutic potential in Alzheimer's disease, this work presents a new approach for identifying proteins that bind to amyloid fibrils. Given that amyloid accumulation is a feature of many other diseases, including Parkinson's disease, Huntington's disease and type 2 diabetes, the approach could have broad therapeutic applications.

---

*Kokkoni et al., 2006*; *Takahashi and Mihara, 2008*; *Cheng et al., 2012*) and small molecules (*Wood et al., 1996*; *Williams et al., 2005*; *McLaurin et al., 2006*; *Necula et al., 2007*; *Bartolini and Andrisano, 2010*; *De Felice et al., 2001*; *Ladiwala et al., 2011*; *Hard and Lendel, 2012*; *Kroth et al., 2012*), that inhibit Aβ fibrillogenesis and/or Aβ-associated cytotoxicity in vitro. While most efforts have targeted the deposition of Aβ fibers as the hallmark of Alzheimer's, smaller amyloid oligomers are now receiving greater attention as the possible toxic entities in Alzheimer's and other neurodegenerative diseases (*Hartley et al., 1999*; *Cleary et al., 2005*; *Silveira et al., 2005*). Furthermore, emerging evidence suggests that mature, end-stage amyloid fibers may serve as a reservoir, prone to releasing toxic oligomer (*Xue et al., 2009*; *Cremades et al., 2012*; *Krishnan et al., 2012*; *Shahnawaz and Soto, 2012*). Recent screens have identified compounds that reduce Aβ cytotoxicity, without interfering with Aβ fibrillation (*Chen et al., 2010*) or promoting the formation of stable Aβ aggregates (*Bieschke et al., 2011*).

Structural information about protein targets often aids drug development, so here we take a structure-based approach, combined with computational screening, to discover amyloid interacting compounds that reduce amyloid toxicity. This approach has been enabled by the determination of atomic structures of the adhesive segments of amyloid fibers, termed steric zippers (*Nelson et al., 2005*), and of solid state NMR-based structures of amyloid fibers (such as full-length Aβ fibers [*Luhrs et al., 2005*; *Petkova et al., 2005*] and the HET-s prion domain complexed with Congo Red [*Schutz et al., 2011*]). The steric zipper structures reveal a common motif for the spine of amyloid fibers, in which a pair of fibrillar β-sheets is held together by the side-chain interdigitation (*Sawaya et al., 2007*). We focus on Aβ, a peptide of 39–42 residues cleaved from the Amyloid precursor protein (APP) associated with Alzheimer's, as a target for inhibitor discovery. The segment Aβ$_{16-21}$ with the sequence KLVFFA is an amyloid-forming peptide, which packs in a steric zipper form, and has been identified as the spine of

the full-length Aβ fiber (*Luhrs et al., 2005*; *Petkova et al., 2006*; *Colletier et al., 2011*). Co-crystal structures have been determined for small molecules in complex with the fibrillar β-sheets of Aβ$_{16–21}$ (*Landau et al., 2011*). One of these structures—Aβ$_{16–21}$ with the dye Orange G—reveals the specific pattern of hydrogen bonds and apolar interactions between orange G and the steric zipper: the negatively charged dye binds specifically to lysine side chains of adjacent sheets, and its planar aromatic portion packs against apolar residues (phenylalanine and valine) of adjacent sheets. By creating a tight, low energy interface across several β-strands within fiber core, this fiber-binding molecule appears to stabilize the fiber structure. With this atomic structure as a basis, we are able to screen for small molecular compounds that bind to amyloid fibers, stabilizing them and possibly reducing amyloid toxicity. Applying our structure-based screening procedure, we screen computationally for compounds that bind to Aβ fibers, termed BAFs (Binders of Amyloid Fibers) and then experimentally test their effects on Aβ aggregation and cytotoxicity.

## Results

### Structure-based screening procedure

We have devised a structure-based procedure for the identification of small molecules that bind to amyloid and affect amyloid toxicity (*Figure 1*). The procedure starts from a co-crystal structure of a ligand bound to an amyloidogenic segment of Aβ (*Landau et al., 2011*), the dye orange G bound to the fiber-like crystal structure of KLVFFA(Aβ$_{16–21}$) segment. This structure reveals the chemical environment or '*pharmacophore*' presented by the ligand binding site of this Aβ segment, that is, orange G binds to stacked β-sheets of Aβ. Knowledge of the amyloid pharmacophore (*Figure 1A*) permitted us to screen for compounds that could be expected to bind in this chemical environment, possibly stabilizing amyloid fibers.

### Construction of compound libraries for computational screening

For assembling the compounds in our screening library, we sought three characteristics: (a) commercially available compounds since we intended to follow the in silico screening with experimental validation; (b) compounds with known three-dimensional structures such that our screening would be as realistic as possible; (c) generally flat compounds able to bind to the β-sheets of the steric zipper, as does orange G. Some ~11,000 compounds having the first two characteristics (CSD-ZINC set) were selected as the intersection of molecules found both in the Cambridge Structure Database (http://www.ccdc.cam.ac.uk) and the Zinc Database of purchasable compounds (http://zinc.docking.org/) (*Irwin and Shoichet, 2005*). This CSD-ZINC set spans a variety of structural shapes and molecular properties. A second set of ~7000 compounds, the Flat Compound Set, was gathered from the ZINC database to include molecules expected to bind to the flat surface of a steric zipper. The members of this set contain multiple aromatic rings or one aromatic ring with additional planar groups.

### Computational screening of compounds that bind to Aβ fibers

Computational screening was carried out with the RosettaLigand program (*Davis and Baker, 2009*), after adapting its docking approach to carry out high-throughput screening (*Figure 2*). The conformational flexibilities of ligand and protein side chains are in a 'near-native' perturbation fashion, meaning that the fine sampling of conformations was restrained to be close to the starting conformation. A balance was achieved between extensive sampling and the speed required for screening a large compound library by fine sampling of side chain and ligand torsion angles only around their starting conformations, as illustrated by sticks in *Figure 2C*.

In the screening steps of computational docking (*Figure 2A*), a library of ~18,000 purchasable compounds (Sets 1 and 2) was scanned computationally for structural compatibility with the pharmacophore (ligand binding site) presented by a single sheet of the Aβ$_{16–21}$ steric zipper. Structural compatibility was assessed by a combination of binding energy (*Meiler and Baker, 2006*) and steric complementarity (*Lawrence and Colman, 1993*). After computational docking, the distribution of calculated binding energies suggests that, statistically the flat compounds from Set 2 fit more snugly on the flat surfaces of Aβ$_{16–21}$ fibers than those with diverse shapes in Set 1 (*Figure 2B*). The best scoring compounds were screened further by requiring that each is also structurally compatible with the solid-state NMR-derived model of the Aβ full-length fiber structure (*Petkova et al., 2006*) (*Figure 1C* and *Figure 1—figure supplement 3*).

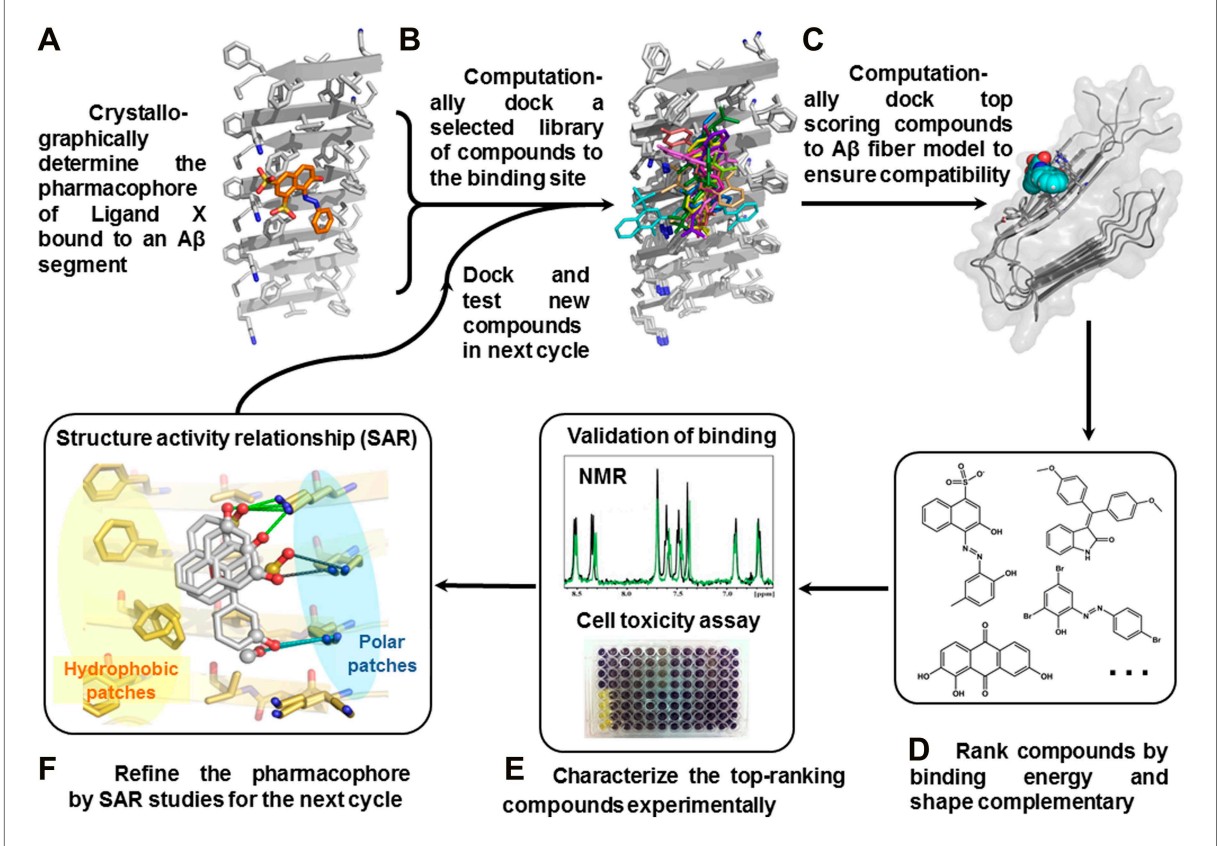

**Figure 1**. Structure-based identification of small compound inhibitors of Aβ toxicity. In step (**A**) the crystal structure (**Landau et al., 2011**) is determined of a complex of an amyloidogenic segment of Aβ (in this case residues 16-KLVFFA-21 of the spine of the Aβ fiber) with an amyloid-binding Ligand X (in this case orange G), revealing aspects of the pharmacophore for Ligand X. Prior to step (**B**) a large library of available compounds is selected for computational docking (~18,000 purchasable compounds in this case). In step (**B**) computational docking is applied to test the compatibility of each member of the library for the pharmacophore of the amyloidogenic segment defined in step (**A**). In step (**C**), the top scoring members of the library are tested for compatibility of binding within a full-length Aβ fiber (in this case the 400 top scoring members were tested on a solid state NMR-derived model of an Aβ fiber, pdb entry 2LMO) (**Petkova et al., 2006**). The representative models from steps **B** and **C** are shown in **Figure 1—figure supplements 1 and 2**. In step (**D**), the compounds are ranked by tightest binding energy and best shape complementarity for the pharmacophore. In step (**E**), the top-ranking compounds (25 in this case) are selected for experimental characterization and validation, including NMR assessment of binding, EM assays of their effects on fiber formation, and cell viability assays for their effects on Aβ cytotoxicity. In step (**F**), new compounds (9 in this case) and compound derivatives (17 in this case) are selected for an additional cycle of computational and experimental testing, based on their similarity to the lead compounds from the initial cycle.

The following figure supplements are available for figure 1:

**Figure supplement 1**. Structural models of the representative BAFs and orange G docked to the side of the KLVFFA(Aβ$_{16-21}$) fiber.

**Figure supplement 2**. Structural models of the representative BAFs and orange G docked onto the full-length Aβ fiber.

**Figure supplement 3**. Alternative binding modes of BAF1 with the Aβ full-length fibers.

## Experimental characterization of BAFs

After in silico screening of a library of ~18,000 purchasable compounds, twenty-five of the top-ranking compounds all with better scores for binding energy and steric complementarity than orange G (**Figure 1D**, **Figure 2—figure supplement 1**), were selected for experimental validation. First these 25 compounds were tested for their ability to protect mammalian cells from Aβ toxicity (**Figure 1E**, **Tables 1 and 2**), and five of them were found to reduce the toxic effects of Aβ. These five were tested for binding to both Aβ$_{1-42}$ and Aβ$_{16-21}$ fibers by NMR. Two were found to have tighter binding than orange G, and the others gave insufficient NMR signals for detection. To expand this set of the five compounds, a

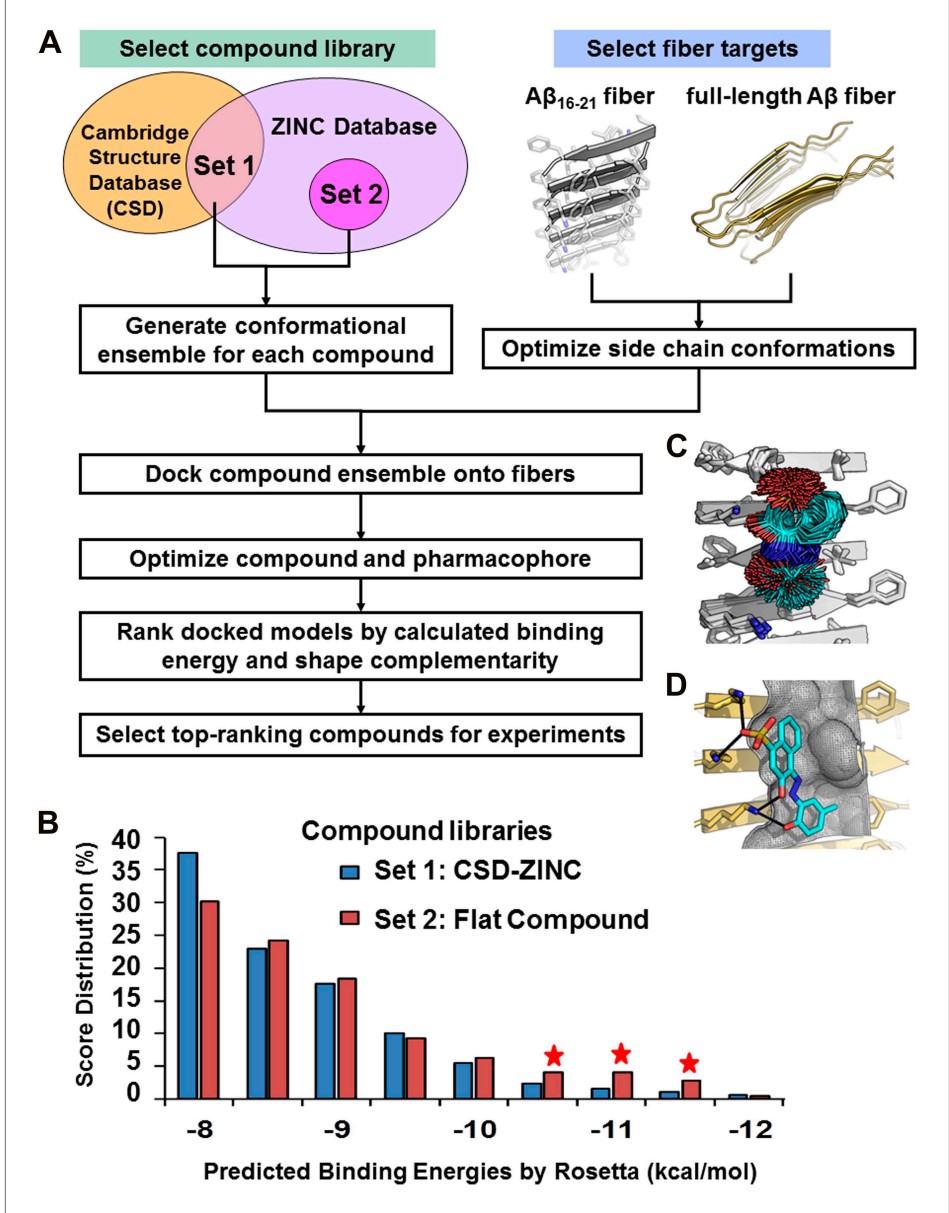

**Figure 2**. Computational screening for fiber-binding compounds. (**A**). Outline of our procedure for structure-based screening. We prepare two sets of compounds (shown in the upper left) for screening against both types of fibers shown in the upper right. Compound Set 1 is the intersection of the ZINC Database of purchasable compounds with the Cambridge structural database (CSD) of known structures. Set 2 consists of other flat aromatic and multiple conjugated compounds found in the ZINC Database. The full description of each computational step is in 'Materials and methods'. (**B**). Distribution of calculated binding energies for the compound libraries of Sets 1 and 2. Those top-ranking compounds have better predicted binding energy than orange G. Structural comparison of docked models of such compound BAF8 and orange-G is discussed in the ***Figure 2—figure supplement 1***. Notice the starred bins which suggest that some members of Set 2, containing flat compounds, tend to be among the top scoring compounds, presumably having the tightest binding to the flat fiber surface. (**C**). The conformational ensemble of a compound representative shown docked onto the Aβ$_{16-21}$ fiber structure. (**D**). A model of BAF8 docked onto an NMR-derived model of full-length Aβ fiber. Notice that the apolar ring structure of the compound binds to the relatively flat apolar (gray) surface of the fiber, and the polar moieties of the compound (red) form hydrogen bonds to the polar groups of the fiber (yellow). The stereo view of BAF8 model is shown in ***Figure 2—figure supplement 2***.

*Figure 2. Continued on next page*

*Figure 2. Continued*

The following figure supplements are available for figure 2:

**Figure supplement 1**. Structural comparison between docked models of BAF8 and orange G.

**Figure supplement 2**. Stereo view of the structural model of BAF8 with Aβ fiber.

second cycle of inhibitor discovery was performed. From the computed positions of the five compounds, a refined pharmacophore was inferred (*Figure 1F*), and used in the next cycle of screening. Added to the compound set were nine additional compounds apparently related to the five lead compounds from the initial cycle, plus 17 chemical derivatives of compounds (*Tables 1 and 3*). The second cycle produced three additional compounds and three compound derivatives that also protected the mammalian cells from Aβ fibers. One of these compounds was confirmed by NMR to bind to Aβ fibers. The detailed description of those experimental results is as follows.

## Inhibition of Aβ$_{1-42}$ toxicity by BAFs

Having identified compounds that bind Aβ fibers, by a structure-based procedure, we tested their effects on the cytotoxicity of Aβ$_{1-42}$ fiber against two mammalian cell lines: PC12 and HeLa (*Figure 3*). Five BAFs—1,4,8,11, and 12—in the initial cycle and three additional BAFs—26, 30, and 31—from the second cycle, with diversified chemical structures shown in *Figure 4*, significantly increased both PC12 and HeLa cell survival after 24 hr incubation with Aβ$_{1-42}$ (0.5 μM) at concentration of 2.5 μM, while the BAFs alone had little or no effect on cell survival (*Figure 3—figure supplement 1*). Three BAFs—11, 26, and 31—showed clear dose-response profiles in their protection of both PC12 and HeLa cells (*Figure 3B*). Among them, the two best BAFs—26 and 31—were tested and did not affect the cytotoxicity of amyloid fibers other than Aβ (*Figure 3—figure supplement 2*). Although all of these BAFs provide protection against Aβ toxicity, none diminish the amount of Aβ fibers in electron micrographs (*Figure 3C*).

## Validation of compound binding by NMR titration

Promising candidate binders from in silico screening and toxicity tests were validated by titration of Aβ fibers into solutions of each compound, as monitored by NMR signals of aromatic protons of the compound (*Figure 5*). The proton resonances of the freely rotating compounds disappear as the compound binds to the fibers. By increasing the amount of fibers, an apparent Kd for compound binding can be estimated. From in silico screening, all tested BAF compounds are calculated to bind more tightly to Aβ fibers than orange G. In NMR studies, the apparent Kd of orange G binding to Aβ$_{16-21}$ fibers was found to be 43 ± 21 μM, whereas the apparent Kd of BAF1 binding to Aβ$_{16-21}$ fibers is 12 ± 7 μM. BAFs were found to bind to both Aβ$_{16-21}$ fibers and Aβ$_{1-42}$ fibers. *Figure 5F* shows a notable correlation between the calculated binding energies and the reduction in NMR peak areas upon Aβ binding. That is, all BAFs with predicted binding energy better than orange G also reduce NMR peak areas more than orange G. On the other hand, BAF31ΔOH, a derivative of BAF31 by removal of a key hydroxyl group essential for binding, exhibits both a worse calculated binding energy and a diminished reduction of NMR peak upon titration of Aβ$_{1-42}$ fibers.

## Structure-activity relationship studies of the Aβ pharmacorphore

Based on the lead compounds found in the initial cycle of the procedure, we carried out a second cycle to expand our understanding of the Aβ pharmacorphore. BAF11 (*Figure 6A*), one of the lead compounds in the initial cycle, was used to perform structure-activity relationship studies. Twelve derivatives of BAF11 were scanned to pinpoint the essential apolar and polar interactions for the pharmacorphore refinement (*Figure 6B*, *Figure 6—figure supplement 1*). These derivatives are grouped in five classes, whose effects on Aβ toxicity have been tested (*Figure 6C*). Classes I and II assess the polar region of BAF11, which makes hydrogen bonds to charged Lys16 ladders of the Aβ fiber: the deletion of the hydroxyl group (Class I) significantly decreased the inhibition of toxicity; the swapping of the hydroxyl group with the aromatic tail (Class II) almost abolished inhibition of toxicity. Classes III, IV, and V focused on the aromatic moieties of BAF11: altering the sizes of aromatic groups (Class III) showed little change in inhibition of toxicity while adding charged or polar groups within

**Table 1.** List of all tested BAF compounds

| Compound | Molecular formula | Molecular weight* | Sources/ purchasing | Rescuing percentage (%) | ZINC entry |
|---|---|---|---|---|---|
| BAF1 | $C_{20}H_8Br_4O_5$ | 648 | Sigma-Aldrich | 44 ± 7 | ZINC04261875 |
| BAF2 | $C_{19}H_{14}O_5S$ | 354 | Sigma-Aldrich | 4 ± 3 | ZINC03860918 |
| BAF3 | $C_{16}H_{13}NO_3$ | 267 | Ryan Scientific | 4 ± 5 | ZINC04289063 |
| BAF4 | $C_{24}H_{16}N_2O_6$ | 428 | Aldrich | 88 ± 22 | ZINC13346907 |
| BAF5 | $C_{16}H_7Na_3O_{10}S_3$ | 524 | Sigma-Aldrich | 11 ± 7 | ZINC03594314 |
| BAF6 | $C_{26}H_{20}N_2$ | 360 | Alfa-Aesar | 5 ± 7 | ZINC08078162 |
| BAF7 | $C_{18}H_{12}N_6$ | 312 | Alfa-Aesar | 2 ± 2 | ZINC00039221 |
| BAF8 | $C_{17}H_{14}N_2O_5S$ | 358 | Sigma-Aldrich | 23 ± 11 | ZINC12358966 |
| BAF9 | $C_{19}H_{13}N_3O_4S$ | 379 | NCI plated 2007† | −3 ± 22 | ZINC03954432 |
| BAF10 | $C_{17}H_{13}NO_3$ | 279 | NCI plated 2007 | 3 ± 5 | ZINC00105108 |
| BAF11 | $C_{20}H_{13}N_2O_5S$ | 393 | NCI plated 2007 | 48 ± 12 | ZINC04521479 |
| BAF12 | $C_{13}H_8Br_3NO$ | 434 | NCI plated 2007 | 38 ± 6 | ZINC12428965 |
| BAF13 | $C_{19}H_{16}ClNO_4$ | 358 | Sigma-Aldrich | 0 ± 2 | ZINC00601283 |
| BAF14 | $C_{10}H_6S_2O_8$ | 318 | Sigma-Aldrich | 3 ± 3 | ZINC01532215 |
| BAF15 | $C_{23}H_{28}O_8$ | 432 | Sigma-Aldrich | 13 ± 4 | ZINC00630328 |
| BAF16 | $C_{19}H_{19}NO_5$ | 341 | Sigma-Aldrich | 5 ± 8 | ZINC28616347 |
| BAF17 | $C_{23}H_{25}N_5O_2$ | 404 | Sigma-Aldrich | 6 ± 3 | ZINC00579168 |
| BAF18 | $C_{24}H_{16}O_2$ | 336 | ChemDiv | 6 ± 2 | ZINC02168932 |
| BAF19 | $C_{18}H_{14}N_2O_6$ | 354 | ChemDiv | 3 ± 4 | ZINC01507439 |
| BAF20 | $C_{25}H_{19}N_5OS$ | 438 | ChemDiv | 8 ± 4 | ZINC15859747 |
| BAF21 | $C_{19}H_{14}Br_2O$ | 418 | ChemDiv | 6 ± 3 | ZINC38206526 |
| BAF22 | $C_{21}H_{16}N_2O_3S_2$ | 408 | Life Chemicals | 3 ± 5 | ZINC04496365 |
| BAF23 | $C_{16}H_{11}ClO_5S$ | 351 | Enamine Ltd | 3 ± 5 | ZINC02649996 |
| BAF24 | $C_{23}H_{19}NO_3$ | 357 | Sigma-Aldrich | 16 ± 5 | ZINC03953119 |
| BAF25 | $C_{14}H_8Cl_2N_4$ | 303 | Sigma-Aldrich | 4 ± 3 | ZINC00403224 |
| BAF26 | $C_{17}H_{10}O_4$ | 278 | Aldrich | 46 ± 23 | ZINC05770717 |
| BAF27 | $C_{21}H_{16}BrN_3O_6$ | 486 | ChemBridge | 4 ± 1 | ZINC01208856 |
| BAF28 | $C_{17}H_{12}N_2O_3$ | 292 | ChemBridge | 2 ± 4 | ZINC00061083 |
| BAF29 | $C_{22}H_{10}N_4O_2$ | 362 | ChemBridge | 1 ± 5 | ZINC00639061 |
| BAF30 | $C_{14}H_8O_5$ | 256 | Aldrich | 18 ± 13 | ZINC03870461 |
| BAF31 | $C_{19}H_{21}NO_3$ | 311 | Sigma | 84 ± 12 | ZINC00011665 |
| BAF32 | $C_{15}H_{14}O_7$ | 306 | Sigma-Aldrich | 15 ± 9 | ZINC03870336 |
| BAF33 | $C_{27}H_{33}N_3O_8$ | 528 | Sigma-Aldrich | 7 ± 2 | SIGMA-R2253§ |
| BAF34 | $C_{30}H_{16}N_4O_{14}S_4$ | 785 | Aldrich | ‡ | ALDRICH-S432830§ |
| orange G | $C_{16}H_{12}N_2O_7S_2$ | 408 | Sigma-Aldrich | −2 ± 8 | ZINC04261935 |

The 25 compounds (BAF1-25) are from the first round, and the nine compounds (BAF26-34) are from the second round. Another set of the 17 derivatives of the BAFs are shown in **Table 3**.

*Molecular weight (anhydrous basis) excluding the salt and water molecules.

†National Cancer Institute (NCI) free compound library (http://dtp.nci.nih.gov/).

‡Toxicity results of BAF34 were not consistent among several independent replica experiments, possibly due to impurity and the high molecular weight of the compound.

§ZINC entry of the compound is not applicable, and the catalog number from Sigma-Aldrich is provided.

aromatic region (Classes IV and V) resulted in a significant decrease of inhibition of toxicity. These differences among BAF11 derivatives in inhibition of toxicity (**Figure 6C**) further validated our structure-based approach and provided guidelines for the refinement of Aβ pharmacophore.

**Table 2.** Detailed list of the active BAF compounds

| Compound | Molecular formula | Molecular weight* | Sources/ companies | Purity | Rescuing percentage§ (%) | | ZINC entry code¶ | SMILES string |
|---|---|---|---|---|---|---|---|---|
| | | | | | PC12 | Hela | | |
| BAF1 | $C_{20}H_8Br_4O_5$ | 647.9 | Sigma-Aldrich | ~99% | 38 ± 11 | 44 ± 7 | ZINC04261875 | c1ccc2c(c1)C(=O)OC23c4ccc(c(c4Oc5c3ccc(c5Br)O)Br)O |
| BAF4 | $C_{24}H_{16}N_2O_6$ | 428.4 | Aldrich | ≥95% | 85 ± 18 | 88 ± 22 | ZINC13346907 | c1cc(c(cc1O)O)c2cc3c(cc2N)oc-4cc(=O)c(cc4n3)c5ccc(cc5O)O |
| BAF8 | $C_{17}H_{14}N_2O_5S$ | 358.4 | Sigma-Aldrich | ≥90% | 26 ± 12 | 23 ± 11 | ZINC12358966 | Cc1ccc(c(c1)/N=N/c2c3ccccc3c(cc2O)S(=O)(=O)[O-])O |
| BAF11 | $C_{20}H_{13}N_2O_5S$ | 393.5 | NCI plated 2007 | † | 51 ± 11 | 48 ± 12 | ZINC04521479 | c1ccc2c(c1)ccc(c2O)/N=N/c3c4ccccc4c(cc3O)S(=O)(=O)[O-] |
| BAF12 | $C_{13}H_8Br_3NO$ | 433.9 | NCI plated 2007 | † | 19 ± 6 | 38 ± 6 | ZINC12428965 | c1cc(ccc1/N=C/c2cc(cc(c2O)Br)Br)Br |
| BAF26 | $C_{17}H_{10}O_4$ | 278.3 | Aldrich | ‡ | 60 ± 21 | 46 ± 23 | ZINC05770717 | c12c(cc(cc1)C(=O)C=O)Cc1c2ccc(c1)C(=O)C=O |
| BAF30 | $C_{14}H_8O_5$ | 256.2 | Aldrich | ‡ | 37 ± 18 | 18 ± 13 | ZINC03870461 | c1cc2c(cc1O)C(=O)c3c(ccc(c3O)O)C2=O |
| BAF31 | $C_{19}H_{21}NO_3$ | 311.4 | Sigma | ≥98% | 92 ± 22 | 84 ± 12 | ZINC03874841 | CCCN1CCC2=C3C1CC4=C(C3=CC(=C2)O)C(=C(C=C4)O)O |

BAFs 1, 4, 8, 11, 12 are from the first round. BAFs 26, 30, 31 are from the second round.

*Molecular weight (anhydrous basis) excluding the salt and water molecules.

†With the standard of NCI free compound library.

‡Analytical data for Aldrich[CPR] products are not available.

§Rescue percentage is a scaled cell survival rate.

¶Entry code for the ZINC database (http://zinc.docking.org).

In the second cycle, nine new compounds were derived from the refined pharmacophore (*Figure 7*). Three of them detoxified Aβ in cell survival assay. BAF31, the best inhibitor which protected mammalian cells from Aβ toxicity in the second cycle, increased cell survival from the 40% induced by Aβ alone to >90% (*Figure 3*). A derivative of BAF31, BAF31ΔOH, lacking the hydroxyl group believed to bind to the Lys residue of the Aβ fiber (shown by the magenta oval in *Figure 8B*), is calculated no longer to bind to the Aβ fiber. NMR and cell viability assessments indicated that BAF31ΔOH binds much less strongly to Aβ fibers than BAF31 itself and shows significantly reduced power to inhibit toxicity (*Figure 8E*). Similarly, the detoxifying profile of derivatives of another inhibitor, BAF30, validated the key interactions of BAF30 across the binding interface (*Figure 9*). Our conclusion is that the NMR binding and toxicity results for the BAF derivatives studied are consistent with our model for the pharmacophore of Aβ (*Figure 10*).

## Discussion

### Structure-based discovery of compounds that bind amyloid fibers

Amyloid fibers differ fundamentally in structure from the enzymes and signaling proteins that are the traditional targets in structure based design of binding compounds, and thus their pharmacophores might be expected to differ fundamentally as would the types of compounds that bind. In general, the binding sites of the traditional targets are often concave pockets; in contrast, the surfaces of amyloid fibers are flat and repetitive along the fiber axis, without well-defined surface cavities. The widely used ligand-docking software, such as DOCK (*Ewing et al., 2001*), or AutoDock (*Morris et al., 2009*), is intended to fit well-defined protein pockets rather than shallow grooves at flat fiber surfaces.

Consequently we have adapted the RosettaLigand program (*Davis and Baker, 2009*) for docking a library of commercially available compounds onto the flat surface of amyloid fibers. Similarly to other software packages, RosettaLigand scores each candidate compound for its energetic fit to its binding site. The initial site is chosen near that occupied by a bound compound, as determined in a crystal structure. The conformational flexibilities of ligand and protein side chains are modeled in a 'near-native' perturbation

**Table 3.** List of the representative BAFs 11, 30, 31 and their derivatives

| Compound | Molecular formula | Molecular weight | Description | Toxicity inhibition (%) | ZINC entry/ catalog no. |
|---|---|---|---|---|---|
| BAF31 | $C_{19}H_{21}NO_3$ | 311 | | 84 ± 12 | ZINC03874841 |
| BAF31ΔOH | $C_{19}H_{21}NO_2$ | 295 | remove one hydroxyl (OH) | 15 ± 2 | ZINC03874841 |
| BAF30 | $C_{14}H_8O_5$ | 256 | | 18 ± 13 | ZINC03870461 |
| BAF30αR | $C_{22}H_{20}O_{13}$ | 492 | add additional R group away from binding interface | 20 ± 10 | ZINC28095922 |
| BAF30σOH$^A$αOH | $C_{14}H_8O_6$ | 272 | change one OH (A) position; add another OH | 9 ± 9 | ZINC03874832 |
| BAF30σOH$^A$ΔOH$^B$αCOO | $C_{15}H_8O_6$ | 284 | move one OH (A) position; delete an OH from loc B; add a carboxyl | 9 ± 3 | ZINC04098704 |
| BAF30σOH$^{AB}$αCH$_3$ | $C_{15}H_{10}O_5$ | 270 | move two OH (AB) positions; add a methyl | 6 ± 3 | ZINC03824868 |
| BAF11 | $C_{20}H_{13}N_2O_5S$ | 393 | | 48 ± 12 | ZINC04521479 |
| BAF11$^{ISO}$ | $C_{20}H_{13}N_2O_5S$ | 393 | isomer form of BAF11 | 33 ± 5 | ZINC12405071 |
| BAF11σR1 | $C_{20}H_{14}N_4O_8S_2$ | 502 | change the aromatic group | 35 ± 9 | ZINC25558261 |
| BAF11σR2 (BAF8) | $C_{17}H_{14}N_2O_5S$ | 358 | change the aromatic group | 22 ± 11 | ZINC12358966 |
| BAF11σR3 | $C_{16}H_{12}N_2O_6S$ | 360 | change the aromatic group | 28 ± 4 | ZINC04900892 |
| BAF11αNO$_2^-$ | $C_{20}H_{12}N_3O_7S$ | 438 | add charged group (nitro) | 15 ± 6 | ZINC16218542 |
| BAF11$^{ISO}$αCOO$^-$ | $C_{21}H_{12}N_2O_7S$ | 436 | BAF11 isomer; add charged group (carboxyl) | 6 ± 5 | ZINC03861030 |
| BAF11$^{ISO}$αSO$_3^-$ | $C_{20}H_{11}N_2O_{11}S_3$ | 552 | BAF11 isomer; add charged group (sulfate) | 2 ± 5 | SIGMA-33936 |
| BAF11ΔOHσR | $C_{20}H_{14}N_2O_4S$ | 378 | remove an OH; change the position of the aromatic group | 15 ± 6 | ZINC04803992 |
| BAF11ΔOHαSO$_3^-$ | $C_{20}H_{14}N_2O_7S_2$ | 458 | remove an OH; add sulfate group | 12 ± 3 | ZINC03954029 |
| BAF11ΔOHαR | $C_{20}H_{18}N_4O_5S$ | 426 | remove an OH; add additional group to the aromatic ring | 12 ± 6 | ZINC04416667 |
| BAF11σOHαR1 | $C_{24}H_{20}N_4O_4S$ | 461 | swap the position of the OH and aromatics | 5 ± 5 | ZINC04804174 |
| BAF11σOHαR2 | $C_{16}H_{19}N_3O_5S$ | 365 | swap the position of the OH and aromatics | 4 ± 6 | ZINC17378758 |

fashion ('Materials and methods'), meaning that the fine sampling of conformations was restrained to be close to the starting conformation. To find the position along the flat fibrillar surface of greatest binding energy for each candidate compound, our screening approach leverages the rotamer repacking algorithm (*Leaver-Fay et al., 2011*) and Rosetta energy function (*Kuhlman and Baker, 2000*) to account for

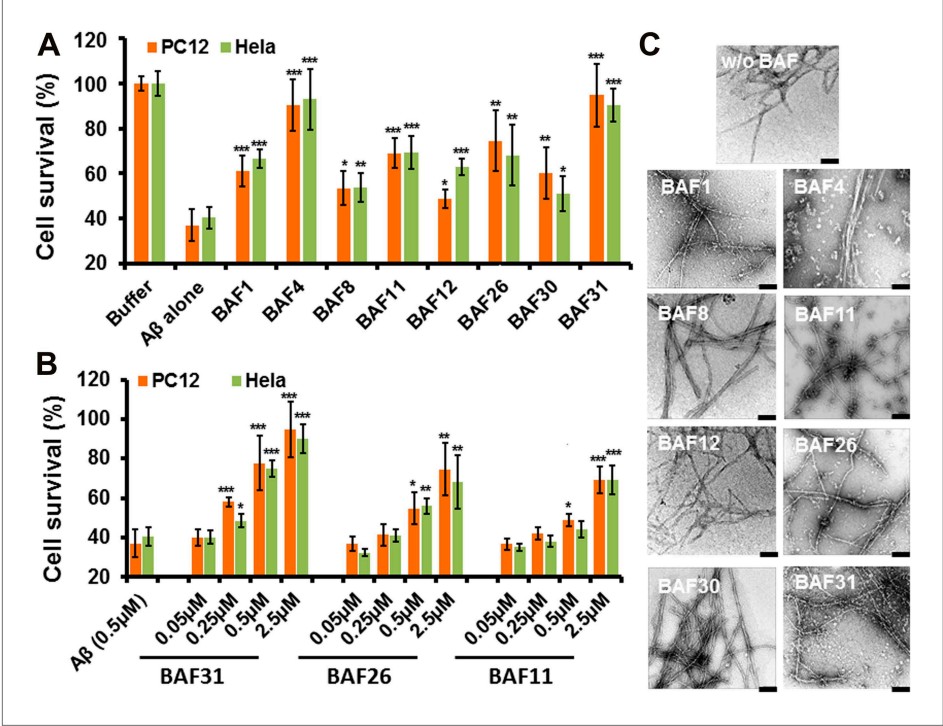

**Figure 3**. Experimental characterization of compounds that bind to amyloid fibers. Our newly discovered BAFs diminish Aβ$_{1-42}$ toxicity without significantly reducing Aβ$_{1-42}$ fibrillation. (**A**). Eight BAFs reduce Aβ toxicity in mammalian cell lines (PC12 in orange; HeLa in green). These identified compounds with diversified chemical structures are quite different from orange G, whose co-crystal structure with an amyloid segment is the basis of our approach (**Figure 4** and **Table 2**). For each compound, 2 to 4 repeats of each independent experiment were performed. For each experimental repeat, four replicates per sample per concentration were tested. The symbol * indicates a p<0.1; the symbol ** indicates a p<0.01 and the symbol *** indicates a p<0.001. The student's *t*-test and p-value analysis are in **Table 4**. (**B**). The representative BAFs—31, 26, and 11—inhibit Aβ cyto-toxicity in a dose-dependent manner. (**C**). Transmission electron microscopy (TEM) images of Aβ fibers alone and Aβ fibers with the BAFs, the same samples prepared for cell viability assay. All 8 BAFs that diminish Aβ toxicity do not noticeably diminish Aβ fibrillation. Scale bars indicate 200 nm.

The following figure supplements are available for figure 3:

**Figure supplement 1**. The BAFs alone exhibit little or no toxicity on mammalian cell lines.

**Figure supplement 2**. BAFs cannot reduce the cytotoxicity of amyloid fibers formed by IAPP and α-synuclein, as much as those fibers formed by Aβ.

flexibility of protein side chains and ligand, which is critical in modeling of such shallow grooves on the fiber surface.

Our procedure identified 34 BAF compounds predicted to bind to Aβ fibers, among which eight BAFs diminish the toxicity of the fibers in mammalian cells. We suggest that the same procedure can be used to discover other compounds that reduce the toxicity of Aβ fibers, starting from other co-crystal structures of Aβ segments with other bound ligands. Similarly, the same procedure can be applied to the discovery of compounds that bind to other amyloid proteins, for use as either toxicity inhibitors or imaging agents for amyloid diagnosis.

## Mechanism of inhibition of Aβ toxicity

Our observation is that our tightest binding BAFs all diminish the toxicity of Aβ fibers, and yet do not substantially diminish the amount of fibers. Further study will be required to understand the molecular mechanism underlying the inhibition of Aβ toxicity, but here we offer the following hypothesis.

**Table 4.** Student's *t*-test and p value analysis suggests that BAFs reduce the cytotoxicity of Aβ fibers significantly

| | Average of cell viability (n = 4) | SD(σ) | Comparison to Aβ fiber alone | |
|---|---|---|---|---|
| | | | *t* value | p value |
| HeLa cell line | | | | |
| Aβ fiber alone | 0.40 | 0.05 | / | / |
| BAF1 | 0.66 | 0.04 | 8.4 | 5E-05 |
| BAF4 | 0.93 | 0.13 | 7.4 | 1E-4 |
| BAF8 | 0.54 | 0.06 | 3.3 | 1E-2 |
| BAF11 | 0.69 | 0.07 | 6.6 | 2E-04 |
| BAF12 | 0.63 | 0.04 | 7.6 | 1E-04 |
| BAF26 | 0.68 | 0.14 | 3.8 | 5E-3 |
| BAF30 | 0.51 | 0.08 | 2.3 | 4E-2 |
| BAF31 | 0.91 | 0.07 | 11.5 | 7E-06 |
| PC12 cell line | | | | |
| Aβ fiber alone | 0.37 | 0.07 | / | / |
| BAF1 | 0.61 | 0.07 | 4.9 | 1E-3 |
| BAF4 | 0.90 | 0.11 | 8.0 | 7E-05 |
| BAF8 | 0.53 | 0.07 | 3.2 | 1E-2 |
| BAF11 | 0.69 | 0.07 | 6.5 | 2E-4 |
| BAF12 | 0.49 | 0.04 | 2.9 | 2E-2 |
| BAF26 | 0.74 | 0.13 | 5.0 | 1E-3 |
| BAF30 | 0.60 | 0.11 | 3.5 | 8E-3 |
| BAF31 | 0.95 | 0.14 | 7.4 | 1E-4 |

The Student's T-test and p-value are based on the comparison to Aβ fiber alone.

Emerging evidence suggests that amyloid oligomers, rather than amyloid fibers, are toxic entities (*Hartley et al., 1999*; *Cleary et al., 2005*; *Silveira et al., 2005*), and that perhaps toxic oligomers can be released from amyloid fibers (*Xue et al., 2009*; *Cremades et al., 2012*; *Krishnan et al., 2012*; *Shahnawaz and Soto, 2012*). By binding to fibers, BAFs stabilize them, thereby shifting the equilibrium of Aβ molecules from smaller, toxic entities towards the fibrillar state. The BAF compounds in their computationally docked sites on Aβ fibers contact several (as few as three and as many as six) adjacent β-strands of the fiber. By creating a low energy binding interface across several fiber strands, the BAFs apparently stabilize the Aβ fibers from breaking into smaller entities.

From previous studies, we expect BAFs to bind to amyloid fibers rather than oligomers. In recent work (*Laganowsky et al., 2012*; *Liu et al., 2012*), we proposed that amyloid forming proteins can enter either of two distinct aggregation pathways, which are separated by an energy barrier. One pathway leads to in-register fibers in which every β-strand lies directly above or below an identical strand in the fiber. The other pathway leads to out-of-register oligomers in which antiparallel β-strands are sheared relative to one another and roll into a β-barrel. We found that three out-of-register amyloid-like structures exhibit cytotoxicity (*Laganowsky et al., 2012*; *Liu et al., 2012*), which tend to be transient, equilibrating eventually into in-register fibers. In our approach, we search for BAFs based on in-register β-sheets rather than out-of-register β-strands found in toxic oligomeric structures, to which our BAFs are not expected to bind (*Figure 11*). We speculate that BAFs stabilize the in-register fibers revealed by our steric zippers, relative to out-of-register toxic oligomers, thereby shifting the equilibrium from toxic oligomers towards fibers (*Figure 12*). Supporting this is our result that diminished toxicity accompanies compound binding.

**Figure 4**. Diversified chemical structures of 8 active BAF compounds that reduce Aβ toxicity. Orange G in an orange box is also displayed for comparison.

## BAFs strengthen the hypothesis that Aβ$_{16-21}$ fibers reflect essential features of full Aβ fibers

The identification of BAFs starts with the atomic structure of orange G bound within the fiber-like crystals of Aβ$_{16-21}$, because as yet there is no high-resolution atomic structure available for ligands bound to full-length Aβ fibers. Nevertheless, we found that BAFs diminish toxicity of full-length Aβ fibers. This finding suggests that the steric zipper structure of Aβ$_{16-21}$ fibers recapitulates some of the essential structural features of full-length Aβ fibers. We are currently attempting cocrystallization of BAFs with Aβ$_{16-21}$ and other steric zipper structures. We speculate that coupled with computational methods, other steric zipper structures could enable the discovery of the lead compounds for inhibitors of other toxic amyloid entities.

## Materials and methods

### Computational procedures

Two choices of compound libraries for structure-based screening

We generated two sets of purchasable compounds to be screened via the computational docking:

1. Cambridge Structure Database (CSD) set. 102,236 organic compounds, whose crystal structures have R-factor better than 0.1, were extracted from the Cambridge Structure Database (version 5.32 November 2010) using ConQuest. The SMILES string of each structure was then used to locate its purchasing information among the ZINC purchasable set (http://zinc.docking.org/) (*Irwin and Shoichet, 2005*) by OpenBabel package (http://openbabel.org/) (*Guha et al., 2006*). The fast index table of all SMILES strings of the ZINC purchasable set was generated to allow the fast search of each CSD structure against ZINC purchasable set. CSD structures that failed in locating their purchasing information (i.e., without any hit in searching against ZINC purchasable set) were omitted. A library of 13,918 structures from CSD representing 11,057 compounds were finally compiled, whose purchasing information is annotated by ZINC purchasable database. The complete list of CSD/ZINC entries of these compounds in this CSD set can be found in *Supplementary file 1*.

2. Flat Compound (FC) set. A library of 6589 compounds containing phenol and less than three freely rotatable bonds were extracted from the ZINC database (http://zinc.docking.org/) (*Irwin and Shoichet, 2005*). Those compounds have a common feature of planar aromatic ring, resulting in a 'flat' compound.

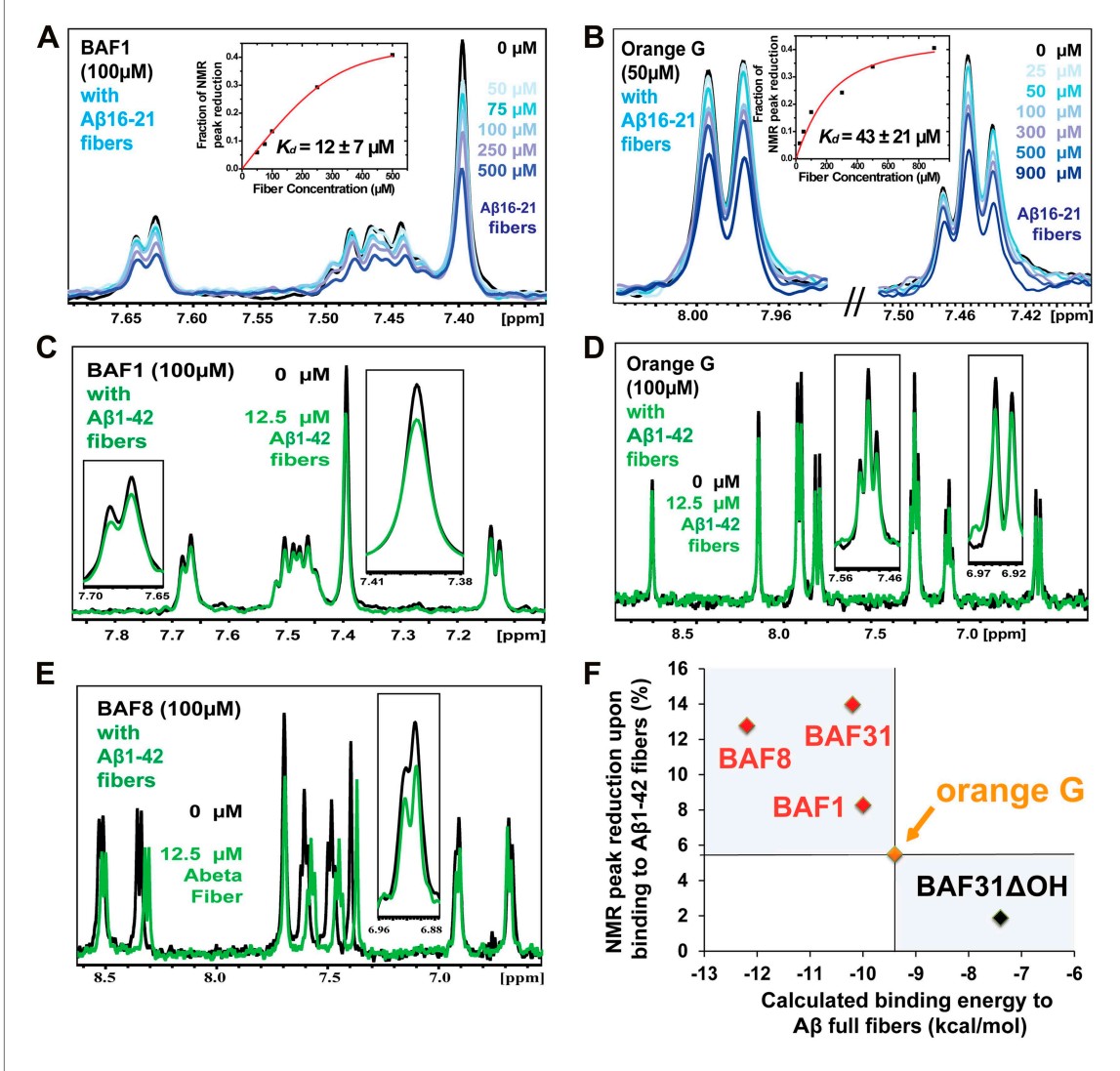

**Figure 5**. NMR evidence for binding of compounds to both Aβ$_{16-21}$ and Aβ$_{1-42}$ fibers. NMR binding experiments were performed on BAF compounds and the dye orange G. By monitoring the aromatic regions of the ${}^1$H NMR spectra of BAFs 1, 8, and 31, these compounds were shown to bind to both Aβ$_{16-21}$ and Aβ$_{1-42}$ fibers more tightly than does orange G. As shown in (**A** and **B**), BAF1 binds to Aβ$_{16-21}$ fibers with affinity stronger than orange G. The determination of binding parameters for Aβ$_{16-21}$ fibers is detailed in ***Table 5*** and ***Figure 5—figure supplements 1 and 3***. In panel (**A**), the ${}^1$H NMR spectrum of compound BAF1 (at 100 μM) is shown as a function of increasing concentration of Aβ$_{16-21}$ fibers (0–500 μM, as monomer). The insert shows the area decrease of BAF1 NMR peaks as a function of Aβ$_{16-21}$ concentration, and the red curve fitting the data defines an apparent Kd of 12 ± 7 μM. In panel (**B**), the NMR spectrum of orange G (50 μM) is plotted against increasing concentration of Aβ$_{16-21}$ fibers (0–950 μM), giving an apparent Kd of 43 ± 21 μM. In (**C**, **D** and **E**), BAFs 1 and 8 both bind to Aβ$_{1-42}$ fibers more strongly than orange G. Notice that the molar ratio of BAFs to Aβ$_{1-42}$ fibers is comparable to that used in cell toxicity assays (***Figure 3***). (**F**). The calculated binding energies of BAFs—1, 8, and 31—to Aβ$_{1-42}$ fibers are compared to the decreases in NMR peak of these compounds upon their binding to full-length Aβ fibers. These three BAFs have higher affinities and a larger NMR peak reduction than orange G while the 'knock-out' derivative with removal of key interactions (BAF31ΔOH) discussed below has a weaker calculated affinity and a smaller NMR peak reduction than orange G. We observe good correlation between computed energies and experimental data from NMR.

The following figure supplements are available for figure 5:

**Figure supplement 1**. NMR peak assignment of BAF1 with Aβ$_{16-21}$ fiber.

**Figure supplement 2**. NMR peak assignment of the control compound orange G with Aβ$_{16-21}$ fiber.

**Figure supplement 3**. NMR titration of BAF8 with Aβ$_{16-21}$ fibers.

**Table 5.** Predicted binding energy and experimental measurement of the binding of two BAFs and orange G against both Aβ$_{16-21}$ (KLVFFA) and full-length Aβ fibers

| | Binding to KLVFFA fiber | | Binding to Aβ fiber | |
| --- | --- | --- | --- | --- |
| | Predicted binding energy (kcal/mol) | NMR Kd (µM) | Predicted binding energy (kcal/mol) | NMR peak reduction (%) |
| BAF1 | −8 | 12 | −10 | 8 |
| BAF8 | −12 | 24 | −12 | 13 |
| orange G | −8 | 43 | −9 | 6 |

The determination of the binding parameters with KLVFFA fiber is detailed in **Table 6**.

The flat compound library includes compounds with similar chemical structures to naturally fiber-binding molecules, for instance, Thioflavin-T (ThT), Congo red, Green tea epigallocatechin-3-gallate (EGCG), and Curcumin. It also includes many natural phenols, such as gallic acid, ferulic acid, coumaric acid, propyl gallate, epicatechin, epigallocatechin, etc. The complete list of ZINC entries of these compounds in this FC set can be found in **Supplementary file 2**.

## Ligand ensemble preparation with near-'native' perturbation

Each molecule in our two compound libraries was prepared for the docking simulations. Hydrogen atoms of each molecule were added for the compounds lacking modeled hydrogens using the program Omega (v. 2.3.2, OpenEye) (**Bostrom et al., 2003**). Ligand atoms were represented by the most similar Rosetta atom type, their coordinates were re-centered to the origin, and their partial charges were assigned by OpenEye's AM1-BCC implementation. We then generated the ligand perturbation ensemble near the crystal conformation (CSD set) or starting conformation (FC set) of each molecule. For each rotatable bond of the ligand, a small degree torsion angle deviation (±5°) was applied. K-mean clustering method was used to generate the ligand perturbation ensemble and similar/redundant conformations (rmsd to the selected conformation is less than 0.5 Å) were omitted. Finally, up to 100 conformations for each ligand were generated and made available for Rosetta LigandDock.

## Rosetta LigandDock with additional near 'native' perturbation sampling

We adopted the docking algorithm based on the method previously described in the RosettaLigand docking paper (**Meiler and Baker, 2006**; **Davis and Baker, 2009**). In general, the algorithm includes three stages: coarse-grained stage, Monte Carlo minimization (MCM) stage and gradient-based minimization stage. Whereas the original RosettaLigand method performed a full sampling of torsional degrees of freedom in the internal ligands and protein side-chains, we made modifications to enable

**Table 6.** Comparison of the measured binding parameters of the representative BAFs with orange G by NMR titrations

| Compound | Predicted binding energy (kcal/mol) | f$_{max}$ | Kd (µM) |
| --- | --- | --- | --- |
| BAF1 | −8 | 0.47 ± 0.04 | 12 ± 7 |
| BAF8 | −12 | 0.82 ± 0.04 | 24 ± 5 |
| Orange-G | −8 | 0.46 ± 0.06 | 43 ± 21 |

The second column lists the predicted binding energy for each top docked model of BAF compounds with KLVFFA fiber, and the binding energy of Orange-G with KLVFFA fiber were also calculated for comparison. Our computational method identified the BAF with better fit to the binding interface than Orange-G. We then used NMR titration to determine the binding affinity. Our previous mass spectrometric analyses of the crystal of the Orange-G with KLVFFA fibers have suggested a binding ratio of compound:fiber with the range of 1:1 to 1:10 (**Landau et al., 2011**). Together with our structural models and single binding site assumption, we estimated the binding ratio to be 1:3. Accordingly, calculated NMR binding parameters are listed in the table. The third column f$_{max}$ is the maximum fraction of NMR signal decrease of compound upon binding saturation ('Materials and methods').

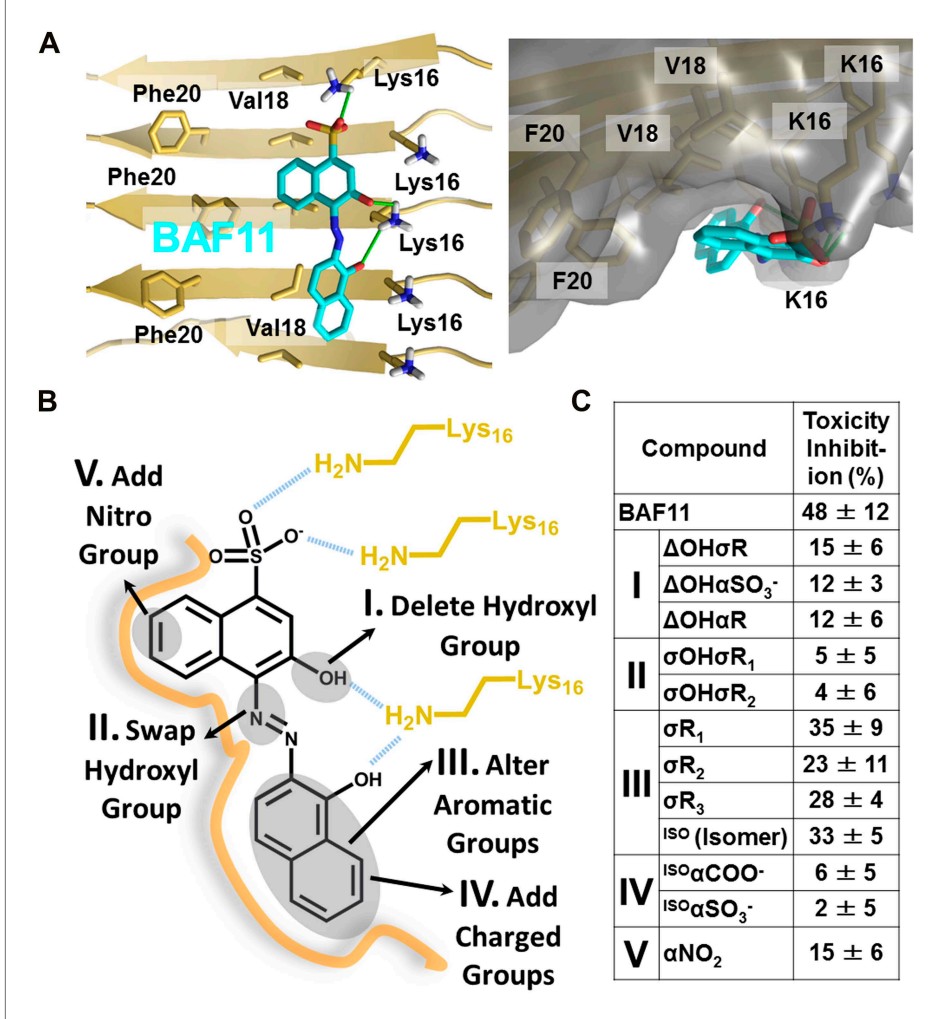

**Figure 6**. Refinement of the Aβ pharmacophore based on studies of BAF11. (**A**) Atomic model of BAF11 from the initial cycle docked on the full-length Aβ fiber, viewed in perpendicular to the fiber axis (left panel) and down the fiber axis (right panel). BAF11 is shown as a cyan stick model, whose polar groups form hydrogen bonds (green thick lines) to Lys16 of Aβ. The extensive non-polar interactions arise from the flat aromatic rings of BAF11 packing against the hydrophobic surface formed by Val18 and Phe20 of Aβ. (**B**) Schematic representation of the polar and nonpolar interactions of BAF11 and its derivatives modeled on the Aβ fiber (in orange and light brown). In the process of the Aβ pharmacophore refinement, five different classes (I–V) of BAF11 derivatives were introduced into the second cycle of screening, to expand the BAF set and to assess the specificity of the compounds identified in the initial cycle. The full description and chemical structure of each derivative are in ***Table 3*** and ***Figure 5—figure supplement 1***. (**C**) Comparison of the toxicity inhibition (defined in 'Materials and methods') among five types of BAF11 derivatives after 24 hr incubation with Aβ (0.5 μM). Notice that all changes to BAF11 which remove binding groups diminish its effectiveness as an inhibitor of toxicity.

The following figure supplements are available for figure 6:

**Figure supplement 1**. Chemical structures of the lead compound BAF11 and its derivatives.

---

the fast run time required by the screening method. Specially, we sampled the ligand and protein side-chain torsion angles in near-'native' perturbation fashion, where only the near-'native' conformation of side-chain and ligand rotamers were allowed and any conformation far away from the starting conformation was omitted. For each protein side-chain, the deviations (±0.33, 0.67, 1 SD) around each input torsion were applied based on the standard deviation value of the same torsion bin from the backbone-dependent Dunbrack rotamer library. For each internal torsional angle of the ligand, the deviations (±5°) around the input torsion were applied as described above.

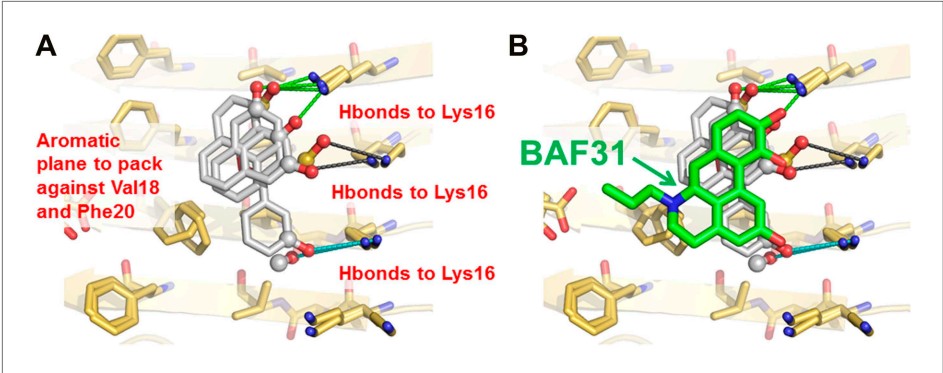

**Figure 7**. New BAFs derived from the refined amyloid pharmacophore. (**A**). Amyloid pharmacophore based on the structural overlay of active BAFs and derivatives. The overlay of the lead compounds from the initial round (BAF4, BAF8, and BAF11) elucidated the consensus of polar and nonpolar interactions at fiber binding interfaces, which sheds light on the amyloid pharmacophore. The amyloid pharmacophore was further refined by iterative approaches of computational docking and experimental testing. The derivatives of those lead compounds were tested to explore the essential role of those consensus interactions, and the differences of binding patterns and toxicity inhibition effects of the BAF derivatives can provide a guideline for the further refinement of amyloid pharmacophore. (**B**). New BAFs were 'designed' based on the refined pharmacophore. One successful example, BAF31 (green sticks) derived from the pharmacophore (grey sticks), showed the enhanced capability of inhibiting Aβ toxicity (**Figure 8C**). The success of developing enhanced binder from pre-defined pharmacophore highlights the important role of iterative docking/test approach in structure-based drug development.

To optimize possible interactions (H-bonding or packing) between compound and fiber, we carried out random perturbations to the TS rigid-body degrees of freedom (5 Å for translational degrees of freedom; 360° for full rotational degrees of freedom) to explore different rigid body arrangements. For each rigid-body perturbation, different conformations of fiber sidechains, and compounds were explored to maximize the binding interactions. We next carried out simultaneous quasi-Newton optimization of the compound rigid body orientation and the sidechain torsion angles, and in some cases, the torsion angles of the compound and the backbone torsion angles in the binding site, using the complete Rosetta energy function.

## Docking of molecules to KLVFFA and Aβ fibrillar structure

The structure of KLVFFA fiber was taken from the co-crystal structure of KLVFFA with orange G (pdb entry: 3OVJ) (**Landau et al., 2011**). After removing orange G, the sidechain torsion of KLVFFA was optimized to correct any conformational bias from the presence of orange G, and then the optimized structure were inspected to ensure that sidechain torsions are still within the original conformation of the co-crystal structure. The Aβ fibrillar structure was from ssNMR fiber structure of full-length Aβ (pdb entry: 2LMO) [40]. The same optimization step was applied before docking. The comparison of docking onto both KLVFFA and Aβ fibrillar structure are discussed in **Figure 13**.

## Post-docking analysis to rank the compounds

The docked compounds were filtered based on the following criteria: (1) The docking models with a compound-fiber van der Waals attractive energy > −7.0 kcal/mol were removed; (2) The docking models with a compound-fiber hydrogen-binding energy >−0.2 kcal/mol were eliminated. The remaining docked compounds were then ranked according to the energy of binding of compound to fiber. We used not only the total binding energy but also on each of the energy components separately (Lennard-Jones interactions, solvation, hydrogen bonding, and electrostatics) (**Lazaridis and Karplus, 1999**; **Kuhlman and Baker, 2000**; **Kortemme et al., 2003**) for ranking. The compounds ranked in the top 40% according to all of these measures were selected. Finally, the compounds were ranked by tightest binding energy (**Meiler and Baker, 2006**) and best shape complementarity (**Lawrence and Colman, 1993**).

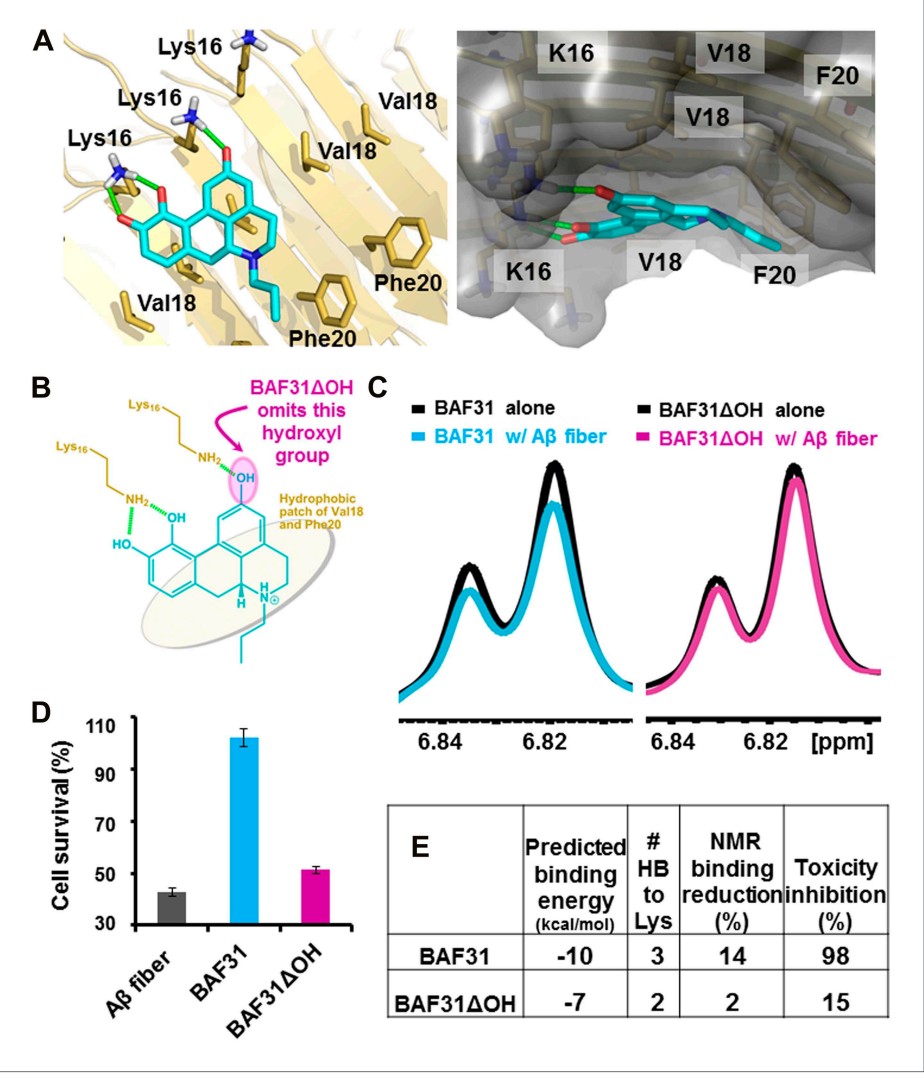

**Figure 8.** Elimination of one key hydrogen bond from BAF31 causes both the loss of NMR binding to Aβ fibers and the decrease in inhibition of Aβ cyto-toxicity. (**A**) Atomic model of the new inhibitor BAF31 (our most tightly binding BAF) derived from the refined pharmacophore (**Figure 7**, **Figure 1F**) in the second cycle, viewed perpendicular to the fiber axis on the left and down the fiber axis on the right. In panel (**B**), one important hydroxyl group forming hydrogen bonds to Lys16 residue of Aβ is highlighted by a magenta circle. (**C**) A representative NMR band (left panel) of mixture of Aβ fiber with the compound BAF31 compares with that (right panel) of Aβ fiber the derivative BAF31ΔOH which omits that important hydroxyl group. Their full NMR spectrums showing the same trend are shown in **Figure 8—figure supplement 1**. (**D**) Cell survival rates after 24 hr incubation with Aβ (0.5 μM), the molar ratio (1:5) of Aβ and the compound is comparable with the ratio in NMR binding experiment (**C**). (**E**) Notably, the elimination of one hydrogen bond from BAF31 (the derivative BAF31ΔOH) causes both the marked decrease in inhibition of Aβ toxicity to HeLa cells (**D**) and the loss of NMR binding to Aβ fibers (**C**).

The following figure supplements are available for figure 8:

**Figure supplement 1.** NMR titration of BAF31 and its derivative with the Aβ$_{1–42}$ fiber.

## Description of geometrical parameters of the interactions between BAFs and Aβ fiber defined based on structure-based screening of Aβ toxicity inhibitor

Based on the rounds of computing search and experimental test, general rules of the essential interactions of BAF binding to Aβ fibers are summarized here. As illustrated in **Figure 10**, the geometrical parameters of those key interactions are specified as followings:

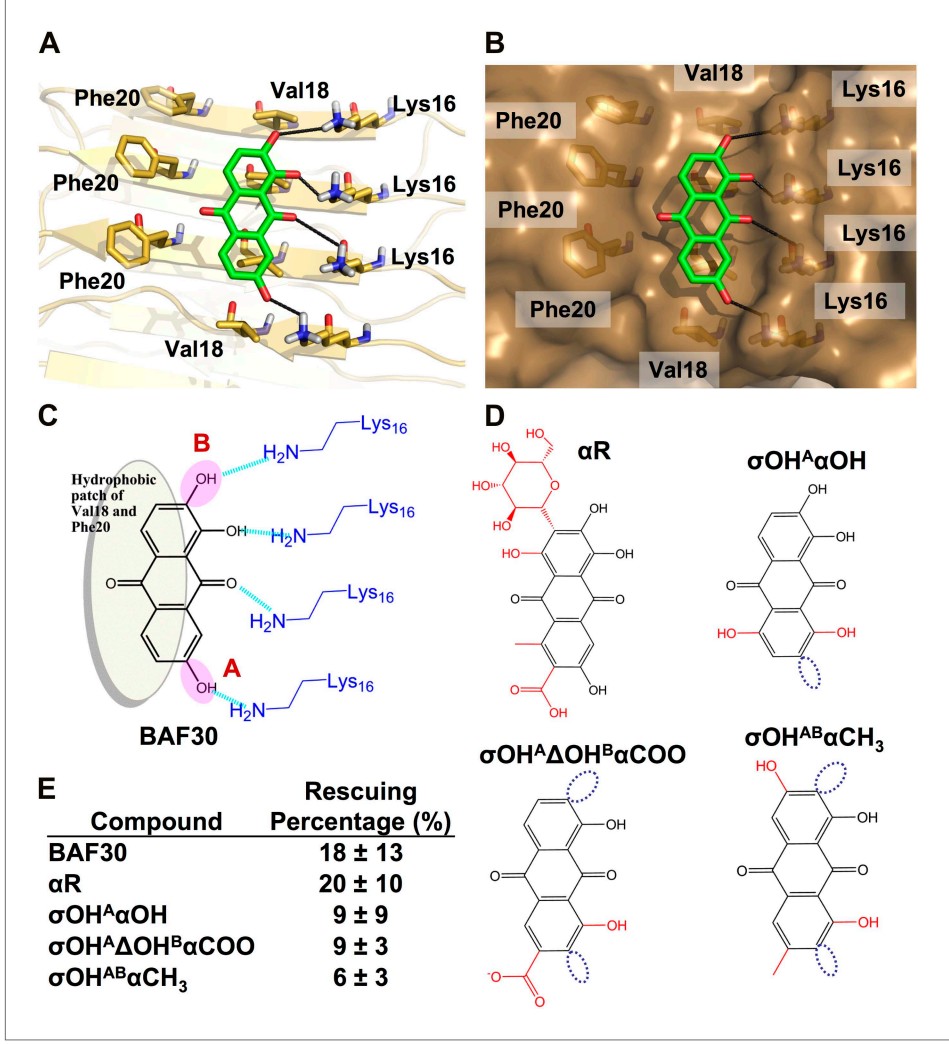

**Figure 9**. Analysis of the lead compound BAF30 and its derivatives. Structural models of BAF30 (green sticks) docked on Aβ fiber structure (in a light yellow color) are shown in (**A** and **B**). The important polar (black hydrogen bonds) interaction between BAF30 and single β-sheet of Aβ fiber, as well as shape complimentary between the aromatic rings of BAF30 and the hydrophobic patches of Aβ fiber are highlighted respectively. Schematic representation of the polar and nonpolar interactions of BAF30 with Aβ fiber is shown in panel (**C**). The magenta circles highlight two important hydroxyl groups which are absent in BAF30 derivatives. (**D**). The chemical structure of each derivative is listed. The dark blue open circles indicate the deletion of the important hydroxyl group. The red color in chemical structures indicates the addition of atoms or groups to BAF30. (**E**). HeLa cell survival rates in the presence of Aβ (0.5 μM monomer equivalent) and BAF30 or the derivatives are compared. The hydrogen bonds between BAF30 and Lys16 residues of Aβ fiber are important for binding of Aβ fiber and inhibition of Aβ toxicity. With additional groups at the opposite side of hydrogen binding sites, the derivative BAF30αR showed little change in toxicity inhibition. However, two BAF30 derivatives (σOH$^A$αOH and σOH$^A$ΔOH$^B$αCOO), which alter or delete the two important hydroxyl groups (magenta circles in panel **C**) of BAF30 that form hydrogen bonds to Lys16, showed a significant decrease in the toxicity inhibition. Furthermore, when BAF30 was modified by shifting both hydroxyl groups (**A** and **B**) to their neighboring positions, the derivative BAF30σOHABαCH$_3$ almost lost the inhibition of Aβ toxicity. The rescuing percentage (%) is defined in 'Materials and methods'.

1. H-bond acceptor (or negative charge) of the inhibitor makes either hydrogen bond or salt bridge to sidechain nitrogen atoms (NZ) of at least two Lysine residues from adjacent Aβ strands along the fiber axis. Our data suggest that the BAFs need to have good contacts across 2 to 4 adjacent Aβ strands, in order to effectively bind to Aβ fiber and reduce Aβ toxicity.

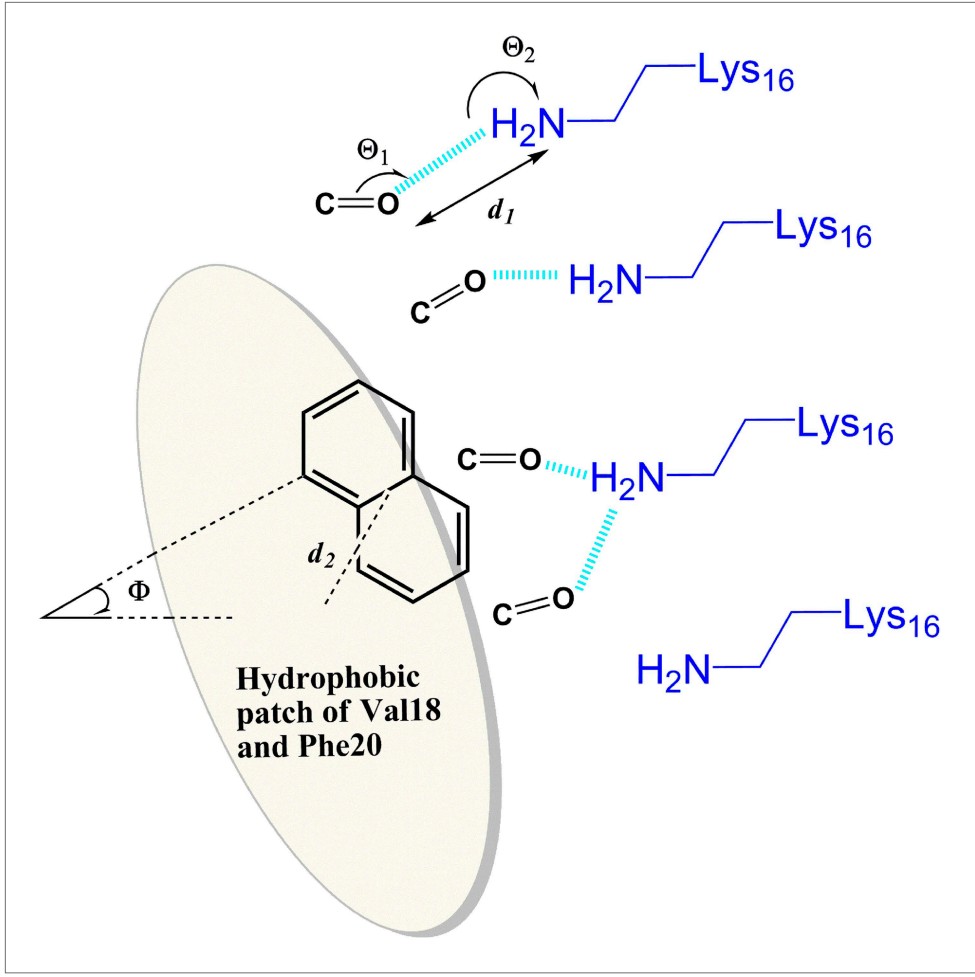

**Figure 10**. General rule of the essential interactions between BAFs and Aβ fiber can be derived from structure-based screening of Aβ toxicity inhibitor. The carbonyl group is used to represent the H-bond acceptor (or negative charge) of BAFs, and the naphthalene ring is used to represent the planar aromatic portion of BAFs. Based on the rounds of computing search and experimental test, the detailed description about essential interactions and geometrical parameters are in 'Materials and methods'.

2. The hydrogen bond or salt bridge described in 1) follows the general rule of H-bond geometry, which are:

   A. Distance ($d_1$, as shown in the figure) between the NZ atom of Lys16 and H-bond acceptor atoms of BAFs: 2.8~3.5 angstrom;
   B. Angle ($\Theta_1$) at BAF H-bond acceptor atoms: 100~150°;
   C. Angle ($\Theta_2$) at the NZ atom of Lys16: 130~180°.

3. Hydrophobic interactions between the apolar residues (phenylalanine18 and valine 20) and the planar aromatic portion of the compounds. The aromatic portion of compounds should be planar or semi-planar to pack against the flat surface of Aβ which spans across at least two adjacent Aβ strands.
4. The hydrophobic interactions described in 3) follow the pi-pi stacking geometry, which are:

   A. Distance ($d_2$) between the center of the apolar sidechains and the center of BAF aromatic rings: 4.0~5.0 angstrom;
   B. Dihedral angle ($\Phi$) between the surface plane defined by Phe18 and Val20 and the aromatic ring of the BAFs: 0~40°.

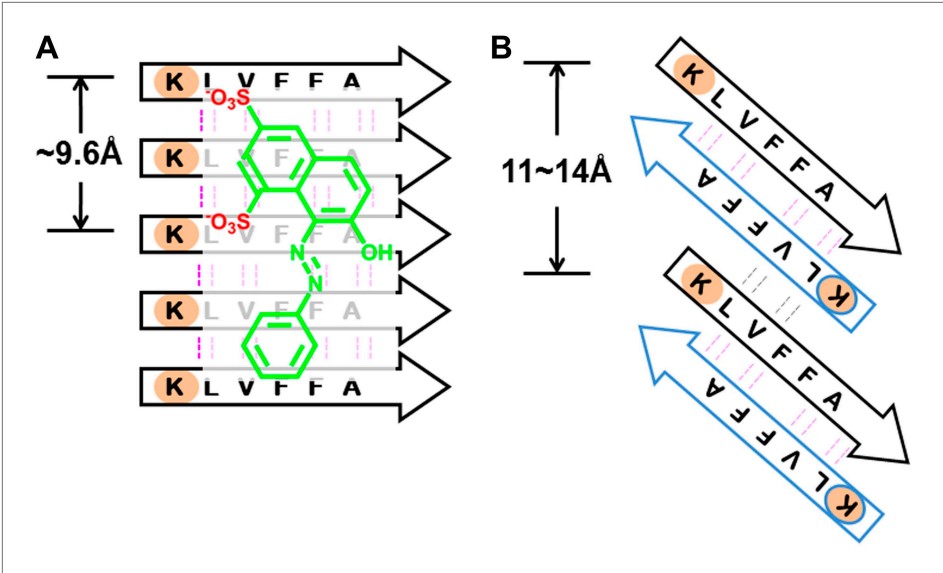

**Figure 11**. BAFs are designed to bind to in-register β-sheets, rather than out-of-register β-sheets. As illustrated in (**A**), BAFs bind to in-register β-sheets. Our structure-based approach searches for BAFs based on in-register β-sheets in Aβ fibers. These BAFs are predicted to bind along the flat hydrophobic surfaces of the fibers and are anchored by polar sidechains of Lysine residues. The Cβ distances between the Lys residues interacting with the BAFs are ~9.6 Å following the stacked arrangement of in-register β-sheets. Orange G, as well as screened BAFs, favorably interact with the in-register fiber and are compatible with the geometry of the Lys residues aligned in in-register β-sheets. As illustrated in (**B**), BAFs cannot bind to out-of-register β-sheets. The estimation of Cβ distance between the lysine residues, based on three out-of-register β-sheets structures previously determined (*Laganowsky et al., 2012*; *Liu et al., 2012*), ranges from 11 Å to 14 Å, quite different from the ~9.6 Å measured in in-register β-sheet. We speculate that the BAFs are unable to bind to out-of-register β-sheets, and this difference accounts for the diminished toxicity that accompanies compound binding. Supporting this is our in vitro cell toxicity tests (**Table 7** and **Figure 11—figure supplement 1**).

The following figure supplements are available for figure 11:

**Figure supplement 1**. Active BAFs show no or little effects on the cyto-toxicity of pre-formed Aβ oligomers.

## Experimental procedures
### Chemicals and reagents
Chemicals were obtained from a variety of companies (*Table 1*) and were of the highest purity available.

### Source of KLVFFA(Aβ$_{16–21}$) and Aβ$_{1–42}$ peptide
N-terminal acetylated and C-terminal amidated KLVFFA(Aβ$_{16–21}$) peptide was synthesized by Celtek Bioscience Peptides (Nashville, TN). Aβ$_{1–42}$ peptide was overexpressed through *Escherichia coli* recombinant expression system and was purified as reported previously (*Finder et al., 2010*). The fusion construct for Aβ$_{1–42}$ expression contains an N-terminal His tags, followed by 19 repeats of Asn-Ala-Asn-Pro, TEV protease site and the human Aβ$_{1–42}$ sequence. Briefly, the fusion construct was expressed into inclusion bodies in *E.coli* BL21(DE3) cells. 8 M urea was used to solubilize the inclusion bodies. Fusion proteins were purified through HisTrap HP Columns, followed by Reversed-phase high-performance liquid chromatography (RP-HPLC). After TEV cleavage, Aβ$_{1–42}$ peptide was purified from the cleavage solution by RP–HPLC followed by lyophilization. To disrupt preformed aggregation, lyophilized Aβ$_{1–42}$ was resuspended in 100% Hexafluoroisopropanol (HFIP) which was finally removed by evaporation.

### Preparation of KLVFFA (Aβ$_{16–21}$) and Aβ$_{1–42}$ fiber samples for 1D ¹H NMR titration measurement
KLVFFA (Aβ$_{16–21}$) peptide was dissolved in PBS buffer, pH 7.4 at the concentration of 1 mM and incubated at 37° with continuing shaking for 3 months. Pre-disaggregated Aβ$_{1–42}$ was dissolved in PBS buffer, pH 7.4 at the concentration of 200 μM and incubated at 37° with continuing shaking for 2 months. For NMR titration

**Table 7.** BAFs reduce Aβ cyto-toxicity by targeting fibers rather than oligomers.

| Compound | Inhibition to the cyto-toxicity of Abeta oligomers (%) | Inhibition to the cyto-toxicity of Abeta fibers (%) |
|---|---|---|
| BAF1 | −4 ± 6 | 36 ± 9 |
| BAF11 | −9 ± 7 | 7 ± 7 |
| BAF26 | −6 ± 6 | 26 ± 7 |
| BAF31 | −17 ± 15 | 58 ± 7 |

The BAF inhibitions of toxicity from either Aβ oligomer or fibers are compared. Four BAFs, which reduce the toxicity of Aβ fibers, show no inhibitory effects to Aβ oligomer toxicity at the equal molar ratio of BAF to Aβ. The inhibition (%) are calculated using the same method defined in 'Materials and methods'. The toxicity assay of Aβ oligomer is described in *Figure 11—figure supplement 1*. The toxicity assay of Aβ fiber is the same as that described in *Figure 3*.

samples preparation, KLVFFA (Aβ$_{16-21}$) or Aβ$_{1-42}$ fiber stocks were diluted in the PBS buffer solution at the indicated concentrations, followed by adding the small molecules from 100 mM stock solutions in DMSO into fibrillar solution. The final concentration of the small molecule was 50uM or 100 μM. The final volume of NMR samples was 500 μL containing 5% D$_2$O. Prior to NMR spectra collection, samples were incubated at room temperature for 0.5 hr. 500 MHz $^1$H NMR spectra were collected on a Bruker DRX500 at 283 K with either 256 or 1024 scans collected depending on the intensity of the small molecule signal. H$_2$O resonance was suppressed via excitation sculpting (*Hwang and Shaka, 1995*); DMSO resonance was suppressed via a frequency shifted presaturation of the DMSO peak. Spectra were processed with XWINNMR 3.6.

## Dissociation constant (Kd) of small molecules to fibers calculated from NMR data

NMR data were analyzed to estimate the binding constant for the interaction between the BAF compounds and KLVFFA fibers. We monitored the decrease in the $^1$H aromatic resonance of the

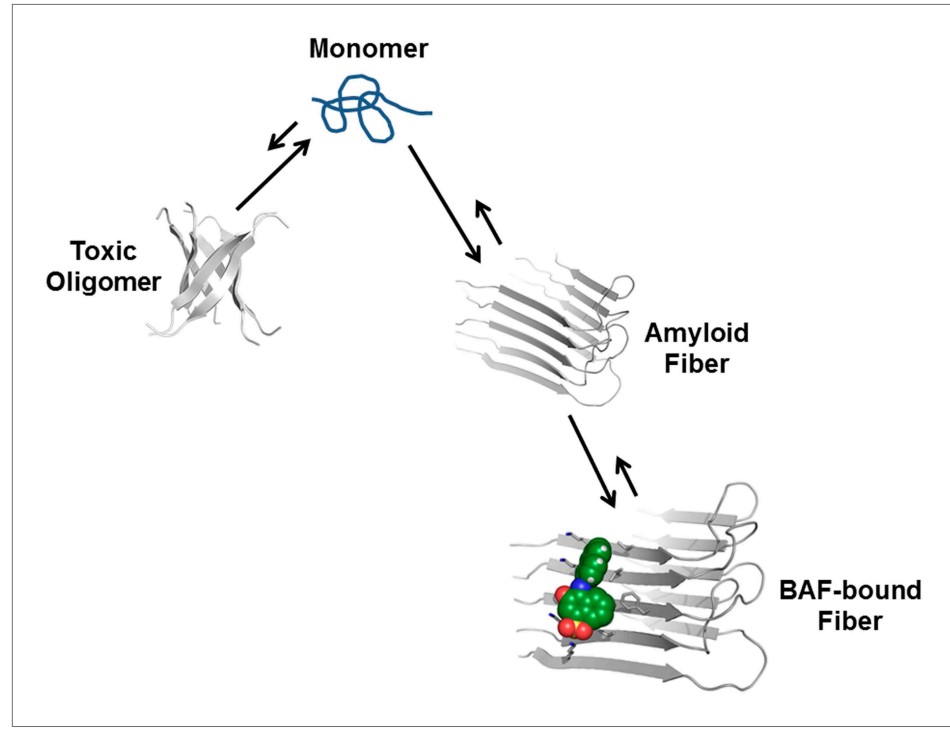

**Figure 12**. Proposed mechanism of how compound binding increases fiber stability and decreases fiber toxicity. BAFs (green) bind to the side of amyloid fibers, stabilizing the fiber, and shifting the equilibrium from smaller and more toxic oligomers towards fibers. This shift in equilibrium reduces amyloid toxicity.

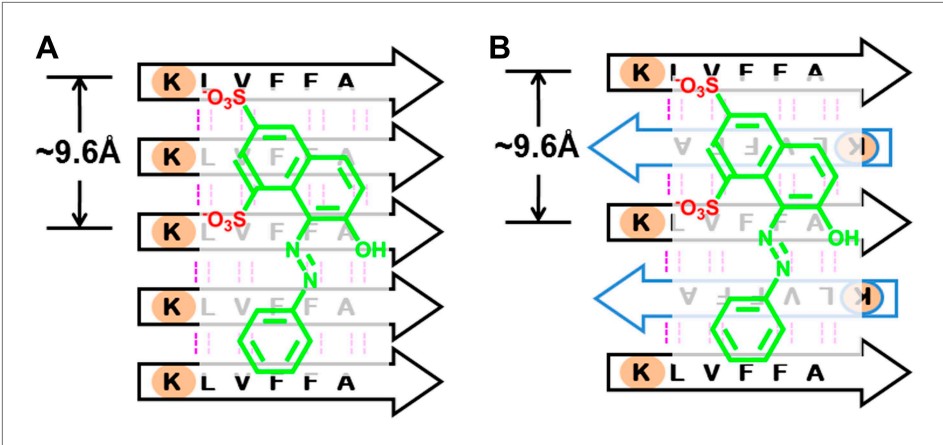

**Figure 13**. BAFs bind to in-register β-sheets and are compatible to both parallel and antiparallel amyloid β-sheets. A subtlety of our procedure for compound discovery is that it involves both parallel (**A**) and antiparallel (**B**) amyloid β-sheets. In the X-ray structure of orange G bound to the segment Aβ$_{16-21}$(KLVFFA) of Aβ, the sheets are antiparallel (**B**). The library of compounds is initially selected based on docking to the antiparallel β-sheet of Aβ$_{16-21}$. In the next step of our procedure, each compound is further screened against the solid-state-NMR-derived model of full-length Aβ fiber, which is a parallel sheet (**A**). The structure models of orange G docked onto Aβ$_{16-21}$ structure and full-length Aβ model are shown in *Figure 13—figure supplement 1*. As simplified here in (**A** and **B**), sulfate ions (red) of orange G are respectively hydrogen bonded to two lysine residues (light brown), which keep nearly identical geometry (the same ~9.6 Å distance between the two lysine residues) in either parallel or antiparallel sheet. Evidence that orange G, as well as BAF compounds identified by our procedure, all bind to both antiparallel and parallel sheets is given by the NMR experiments summarized in *Figure 5*, where orange G and BAFs are shown to bind to both Aβ$_{16-21}$ and full-length Aβ fibers. Apparently both parallel and antiparallel amyloid β-sheets are effective in binding to the same compounds.

The following figure supplements are available for figure 13:

**Figure supplement 1**. Structural models of orange G docked onto the antiparallel Aβ$_{16-21}$ (A) and parallel full-length Aβ (B) fiber.

compounds as a function of increasing concentrations of KLVFFA fibers. The general equation for deriving the apparent dissociation constant (Kd) is as follows:

For a general reaction of a ligand binding to fibers (containing N monomers):

$$F(ibril)_N + L(igand) \leftrightarrow F_N L.$$

We estimated the concentration of fibers at any given monomer concentration as:

$$[F(iber)N] = [Fmomomer] * (1 \text{ fiber/N monomers}),$$

and then we could get:

$[F_N] = \dfrac{[F]_T}{N} - [F_N L]$, $[L] = [L]_T - [F_N L]$, where [F]$_T$ is the total monomer concentration, [L]$_T$ is the total ligand concentration and [F$_N$L] is the concentration of bound fiber;

$$K_d = \frac{[F_N][L]}{[F_N L]} \text{ and } K_d = \frac{\left(\dfrac{[F]_T}{N} - [F_N L]\right)([L]_T - [F_N L])}{[F_N L]},$$

and thus

$$[F_N L]^2 - \left(\frac{[F]_T}{N} + [L]_T + K_d\right)[F_N L] + \frac{[F]_T [L]_T}{N} = 0.$$

Finally, we could get the concentration of bound complex $[F_N L]$:

$$[F_N L] = \frac{\left(\frac{[F]_T}{N} + [L]_T + K_d\right) - \sqrt{\left(\frac{[F]_T}{N} + [L]_T + K_d\right)^2 - \frac{4[F]_T[L]_T}{N}}}{2} \quad (1)$$

We then applied this *equation (1)* to our NMR experiments, where we monitored the integrated area of each NMR peak ($A$) of the compounds over a range of KLVFFA fiber concentrations. Assuming the complex of the BAF compound with fiber is in fast exchange, the peak area is the average of the peak signals for free and bound states, weighted by the fraction of the observed molecule in each state:

$$A = f_L A_L + f_{F_N L} A_{F_N L}.$$

And the change in NMR peak area ($\mathbf{\Delta A}$),

$$\Delta A = A_L - A = f_{F_N L}(A_L - A_{F_N L})$$

$$\frac{\Delta A}{(A_L - A_{F_N L})} = \frac{\Delta A}{\Delta A_{max}} = \frac{\Delta A/A_L}{\Delta A_{max}/A_L} = \frac{\%_{\Delta A}}{\%_{\Delta A_{max}}} = f_{F_N L} = \frac{[F_N L]}{[L]_T}$$

$$\Delta A/A_L = \Delta A_{max}/A_L \frac{\left(\frac{[F]_T}{N} + [L]_T + K_d\right) - \sqrt{\left(\frac{[F]_T}{N} + [L]_T + K_d\right)^2 - \frac{4[F]_T[L]_T}{N}}}{2[L]_T}.$$

Hence, the observed fraction of peak area change during the titration of increasing fiber concentration against fixed small compound,

$$f_{obs} = f_{max} \frac{\left(\frac{[F]_T}{N} + [L]_T + K_d\right) - \sqrt{\left(\frac{[F]_T}{N} + [L]_T + K_d\right)^2 - \frac{4[F]_T[L]_T}{N}}}{2[L]_T}.$$

Our structural model suggests that one BAF compound binds three fiber monomers. To obtain the Kd, we fit the equation for 1:3 (small molecule:fiber) binding to the NMR titration curve (N = 3), with $f_{obs}$ defined as the fraction of peak area decrease $\left(\frac{\Delta A}{A_L}\right)$ for each titration experiment, and $f_{max}$ defined as the fraction maximum of peak area decrease $\left(\frac{A_{max}}{A_L}\right)$ for the saturated complex.

## MTT cell viability assay

We performed MTT-based cell viability assay to assess the cytotoxicity of A$\beta_{1-42}$ with or without the addition of BAFs and orange G. A CellTiter 96 aqueous non-radioactive cell proliferation assay kit (MTT) (Promega cat. #G4100, Madison, WI) was used. HeLa and PC-12 (ATCC; cat. # CRL-1721, Manassas, VA) cell lines were used for measuring the toxicity of A$\beta_{1-42}$. Prior to toxicity test, both HeLa and PC-12 cell lines were plated at 10,000 cells per well in 96-well plates (Costar cat. # 3596, Washington, DC). HeLa cells were cultured in DMEM medium with 10% fetal bovine serum, PC-12 cells were cultured in ATCC-formulated RPMI 1640 medium (ATCC; cat.# 30–2001) with 10% heat-inactivated horse serum and 5% fetal bovine serum. Cells were cultured in 96-well plates for 20 hr at 37°C in 5% CO$_2$. For A$\beta_{1-42}$ and BAFs samples preparation, purified A$\beta_{1-42}$ was dissolved in PBS at the final concentration of 5 μM, followed by the addition of BAFs at indicated concentrations. The mixtures were filtered with a 0.2-μm filter and further incubated for 16 hr at 37°C without shaking for fiber formation. To start the MTT assay, 10 μl of pre-incubated mixture was added to each well containing 90 μl medium. After 24 hr incubation at 37°C in 5% CO$_2$, 15 μl Dye solution (Promega cat. #G4102) was added into each well. After incubation for 4 hr at 37°C, 100 μl solubilization Solution/Stop Mix (Promega cat. #G4101) was added to each well. After 12 hr incubation at room temperature, the absorbance was measured at 570 nm with background absorbance recorded at 700 nm. Four replicates were measured for each of the samples. The MTT cell viability assay measured the percentage of survival cell upon the

treatment of the mixture of $A\beta_{1-42}$ and BAFs. The toxicity inhibition (%) or rescuing percentage (%) of each BAF compound was calculated by normalizing the cell survival rate using the PBS buffer-treated cells as 100% and 0.5 µM (final concentration) $A\beta_{1-42}$ fiber alone-treated cell as 0% viability.

## Transmission electron microscopy (TEM)

TEM was performed to visualize the fibrillation of $A\beta_{1-42}$ in presence of BAFs. The samples of $A\beta_{1-42}$ and BAFs mixture for TEM measurement were the same as those for MTT assay. For specimen preparation, 5 µl solution was spotted onto freshly glow-discharged carbon-coated electron microscopy grids (Ted Pella, Redding, CA). Grids were rinsed twice with 5 µl distilled water after 3 min incubation, followed by staining with 1% uranyl acetate for 1 min. A CM120 electron microscope at an accelerating voltage of 120 kV was used to examine the specimens. Images were recorded digitally by TIETZ F224HD CCD camera.

## ThT fibrillation assay

Purified $A\beta_{1-42}$ was dissolved in 10 mM NaOH at the concentration of 200 µM, followed by sonication for further solubilizing $A\beta_{1-42}$. $A\beta_{1-42}$ was diluted into PBS buffer at the final concentration of 20 µM, and was mixed with 20 µM Thioflavin T (ThT) and different concentrations of BAFs. The reaction mixture was filtered with a 0.2 µm filter, split into four replicates and placed in a 96-well plate (black with flat optic bottom). The ThT fluorescence signal was measured every 5 min using the Varioskan plate reader (Thermo Fisher Scientific, Inc) with excitation and emission wavelengths of 444 and 484 nm, respectively, at 37°C.

## Acknowledgements

We thank N Wu for help with initial computational work, R Peterson and E Hartman for help with NMR experiments, A Soragni, MR Sawaya, H Chang, D Li, D Anderson, P Bajaj, J Bowie, T Yeates, and F Guo for discussion, and HHMI, NIH, NSF, and DOE for support.

## Additional information

### Funding

| Funder | Grant reference number | Author |
| --- | --- | --- |
| National Institutes of Health | AG029430 | Lin Jiang, Cong Liu, David S Eisenberg |
| US Department of Energy | DE-FC02-02ER63421 | David S Eisenberg |
| Howard Hughes Medical Institute | | Lin Jiang, David S Eisenberg |

The funders had no role in study design, data collection and interpretation, or the decision to submit the work for publication.

### Author contributions

LJ, Conceived and designed the projects; Created structure-based screening protocols; Identified the compounds and analyzed the data; Contributed to the crystallographic efforts; Wrote the manuscript coordinating contributions from other authors; CL, Conceived and designed the projects; Performed the EM and cell viability assays; Contributed to the crystallographic efforts; DL, Performed NMR studies; ML, MZ, MPH, Contributed to the crystallographic efforts; DSE, Conceived and designed the projects; Wrote the manuscript coordinating contributions from other authors

## Additional files

### Supplementary files

• Supplementary file 1. Compound Library Set 1: Cambridge Structure Database (CSD) set.

• Supplementary file 2. Compound Library Set 2: Flat Compound (FC) set.

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
