## [Decision Letter]

Thank you for sending your work entitled “Structure-based discovery of fiber-binding compounds that reduce the cytotoxicity of amyloid beta” for consideration at *eLife*. Your article has been favorably peer reviewed by two reviewers, one of whom is a Senior editor.

The Senior editor has assembled the following comments to help you prepare a revised submission in which you should respond to the issues raised by the reviewers.

In this paper, David Eisenberg and colleagues use computational docking to discover small molecules that bind to and stabilize a variety of peptide fibers formed by fragments of amylogenic proteins, and in this way suppress the toxicity of amyloid fibrils of the Alzheimer’s Abeta peptide. They validate the binding of the peptides by NMR, followed up by assays for fibril formation, and also show that the small molecules alleviate the toxicity of the amylogenic peptides to mammalian cells. This work builds on earlier results from the Eisenberg group that established high-resolution crystal structures for many different kinds of peptide fibrils and revealed how small molecules such as Orange G bind to them. The work also follows on earlier studies from others (for example, Pastore and Wanker) on the effects of small molecules on amylogenic peptides. What is new in the present work, however, is the use of computational docking to come up with novel pharmacophores. This is interesting because the binding sites on the fibrils are quite different from the deep hydrophobic cavities that are the targets of conventional docking efforts. Here, the binding sites are flat, and although hydrophobicity is still important, the small molecules lie along the flat surfaces of the fibrils and are anchored by polar residues in a way that stabilizes the fibrils.

The paper is generally clearly written and illustrated. The great interest in developing drugs against amyloid diseases as well as the creative use of computational methods towards that end makes this paper suitable for publication in *eLife*.

Major comment:

A stronger case would be made if the authors tested whether their small molecule compounds bind to and reduce toxicity of oligomers. This would be a good test of their hypothesis that their compounds target fibrils (not oligomers) and prevent toxicity by reducing the release of toxic oligomers from fibrils (i.e., shifting the equilibrium toward fibrils). Do the authors have data that can address this question, or can such data be obtained relatively easily?

---

## [Author Response]

*A stronger case would be made if the authors tested whether their small molecule compounds bind to and reduce toxicity of oligomers. This would be a good test of their hypothesis that their compounds target fibrils (not oligomers) and prevent toxicity by reducing the release of toxic oligomers from fibrils (i.e., shifting the equilibrium toward fibrils). Do the authors have data that can address this question, or can such data be obtained relatively easily*?

We thank the reviewers for raising this important question. To explore the possibility raised by the reviewers that our small molecule compounds (termed BAFs) might target oligomers, we designed experiments to test whether BAFs are capable of inhibiting pre-formed toxic Abeta oligomers using the MTT cell viability assay. The results of these new experiments are given in new Table 7 and Figure 11—figure supplement 1: four BAFs, which significantly reduce Abeta toxicity, show no protective effects on the toxicity of pre-formed Abeta oligomers. These data verify that BAFs detoxify Abeta42 *not* by targeting toxic oligomers, as requested by the reviewers.

Also there are theoretical reasons for expecting that compounds designed to bind fibers will not bind oligomers. These reasons are based on recent work (Laganowsky et al., Science 2012; Liu et al., PNAS 2012), which shows that the structural features of a toxic oligomer are distinct from those observed in fibrils. As we detail in the Discussion section of the present paper, we applied our structure-based approach to search for BAFs that bind to flat in-register β-sheets in fibril structures rather than out-of-register β-strands found in toxic oligomeric structures. As highlighted in Figure 11, these small molecules are predicted to bind along the hydrophobic in-register β-sheet surfaces of the fibrils and to be anchored by polar residues (i.e., Lysine 16 of Abeta fibril in this case). The distances between the Lys residues that interact with the small molecules have distinct differences between in-register β-sheets (left panel) and out-of-register β-strands (right panel). Since we search for the small molecules (in a green color) compatible with the geometry of Lys residues in in-register β-sheets, these small molecules are not expected to bind out-of-register oligomer. Together with our new experimental results, we speculate that the BAFs are unable to bind to Abeta oligomers, and this difference accounts for the diminished toxicity that accompanies specific binding of the small molecules.